# Targeting the BRD4/FOXO3a/CDK6 axis sensitizes AKT inhibition in luminal breast cancer

Jingyi Liu[1,2,3], Zhibing Duan 🅳 [1,2], Weijie Guo[1,2], Lei Zeng[4,5], Yadi Wu[2,6], Yule Chen[1,2], Fang Tai[7,8,9], Yifan Wang[1,2], Yiwei Lin 🅳 [1,2], Qiang Zhang[4,5], Yanling He[7,8,9], Jiong Deng[10], Rachel L. Stewart[2], Chi Wang[2], Pengnian Charles Lin[3], Saghi Ghaffari[11], B. Mark Evers[2,12], Suling Liu[13], Ming-Ming Zhou[4], Binhua P. Zhou[1,2] & Jian Shi[1,2,7,8,9]

BRD4 assembles transcriptional machinery at gene super-enhancer regions and governs the expression of genes that are critical for cancer progression. However, it remains unclear whether BRD4-mediated gene transcription is required for tumor cells to develop drug resistance. Our data show that prolonged treatment of luminal breast cancer cells with AKT inhibitors induces FOXO3a dephosphorylation, nuclear translocation, and disrupts its association with SirT6, eventually leading to FOXO3a acetylation as well as BRD4 recognition. Acetylated FOXO3a recognizes the BD2 domain of BRD4, recruits the BRD4/RNAPII complex to the CDK6 gene promoter, and induces its transcription. Pharmacological inhibition of either BRD4/FOXO3a association or CDK6 significantly overcomes the resistance of luminal breast cancer cells to AKT inhibitors in vitro and in vivo. Our study reports the involvement of BRD4/FOXO3a/CDK6 axis in AKTi resistance and provides potential therapeutic strategies for treating resistant breast cancer.

[1] Department of Molecular and Cellular Biochemistry, University of Kentucky College of Medicine, Lexington, KY 40506, USA. [2] Markey Cancer Center, University of Kentucky College of Medicine, Lexington, KY 40506, USA. [3] Center for Cancer Research, National Cancer Institute-Frederick, Frederick, MD 21702, USA. [4] Department of Pharmacological Sciences, Icahn School of Medicine at Mount Sinai, New York, NY 10029, USA. [5] Bethune Institute of Epigenetic Medicine, The First Hospital, Jilin University, Changchun, Jilin 130021, China. [6] Department of Molecular and Biomedical Pharmacology, University of Kentucky College of Medicine, Lexington, KY 40506, USA. [7] Department of Pathology, School of Basic Medical Sciences, Southern Medical University, Guangzhou, Guangdong 510515, China. [8] Department of Pathology, Nanfang Hospital, Southern Medical University, Guangzhou, Guangdong 510515, China. [9] Guangdong Provincial Key Laboratory of Molecular Tumor Pathology, Southern Medical University, Guangzhou, Guangdong 510515, China. [10] Key Laboratory of Cell Differentiation and Apoptosis of Chinese Ministry of Education, Shanghai Jiao Tong University School of Medicine, Shanghai 200025, China. [11] Department of Developmental and Regenerative Biology, Icahn School of Medicine at Mount Sinai, New York, NY 10029, USA. [12] Department of Surgery, University of Kentucky College of Medicine, Lexington, KY 40506, USA. [13] Key Laboratory of Breast Cancer in Shanghai, Department of Breast Surgery, Cancer Institute, Fudan University Shanghai Cancer Center, Shanghai 200032, China. Correspondence and requests for materials should be addressed to S.L. (email: suling@fudan.edu.cn) or to M.-M.Z. (email: ming-ming.zhou@mssm.edu) or to B.P.Z. (email: peter.zhou@uky.edu) or to J.S. (email: jianshi@smu.edu.cn)

Breast cancer is a heterogeneous disease[1,2], characterized into at least four different subtypes: luminal A, luminal B, ERBB2 overexpression, and basal-like[3,4]. Mutations of *PIK3CA* gene, which encodes the catalytic subunit (p110α) of PI3Kα, occur in almost 40% of ER$^+$/luminal subtype. In addition, mutations of *PTEN* and *INPP4B* contribute to activation of the phosphatidyl inositol 3-kinase (PI3K)/AKT pathway in this subtype[5]. The PI3K/AKT pathway has key roles in regulating growth, survival, and metabolism in both normal and malignant cells. For example, AKT inhibits Forkhead box O (FOXO)-induced expression of growth inhibition and apoptosis induction genes by phosphorylating FOXOs and blocking their nuclear translocation[6,7]. These findings indicate that activation of the PI3K/AKT pathway is likely a causal genetic event in the luminal subtype of breast cancer; thereby, inhibition of this pathway represents a top priority for therapeutic intervention. Indeed, numerous clinical trials have evaluated the efficacy of over 30 drugs targeting different steps of the PI3K/AKT pathway in breast and other cancers, including several AKT inhibitors (AKTis) such as MK2206, AZD5363, and GSK690693. Although these inhibitors have shown evidence of suppressing growth and inducing apoptosis of luminal breast cancer cells, responses of solid tumors to monotherapy have been modest and accompanied by rapid emergence of drug resistance. For example, AKT inhibition induces the expression and phosphorylation of HER3, IGF-1R, and insulin receptor through FOXO-dependent transcriptional activation, suggesting that targeting different nodes of feedback regulation of PI3K/AKT inhibition may improve the killing effects of these inhibitors. Intriguingly, FOXOs proteins are usually deemed as tumor suppressors because of their growth-inhibitory and cell death-inducing ability; the functional roles and downstream target genes of FOXOs involved in drug resistance remain obscure.

The Cancer Genome Atlas (TCGA) data also indicate frequent amplification of *cyclin D1* (40%) and low levels of *RB1* mutations in luminal-type breast cancer[5]. The cyclin D1/CDK4/6 complex phosphorylates the retinoblastoma (Rb) protein, which leads to cell cycle activation[8]. Results from several studies indicate that *CDK4* and *CDK6* have an important role in estrogen-stimulated proliferation of breast cancer cells in early to mid G$_1$ phase[9,10]. Thus, CDK4 and CDK6 represent valuable therapeutic targets of ER$^+$ advanced breast cancer. Consistent with this idea, combination of a CDK4/6 inhibitor with an aromatase inhibitor achieves significant effect on suppressing advanced ER + /luminal subtype of breast cancer[11]. In addition, a combinatorial drug screen on multiple PIK3CA mutant cancers with decreased sensitivity to PI3K inhibitors revealed that combined CDK4/6-PI3K inhibition synergistically reduces cell viability[12]. Although the combination of PI3K and CDK4/6 inhibitors overcomes intrinsic and adaptive resistance leading to tumor regressions in PIK3CA mutant xenografts, the molecular mechanism underlying the resistance of AKTi and the synergy seen on the PI3K inhibitors and CDK4/6 inhibitors remain elusive.

Recently, inhibitors of BRD4, a BET (bromodomain and extra-terminal domain) family member, have shown significant effects in hindering tumor growth by suppressing the expression of oncogenes[13,14]. BRD4 has the capacity to assemble diverse transcriptional complexes on gene super-enhancers and activate RNA polymerase II-dependent transcriptional elongation. In the later, BRD4 was found to preferentially occupy on oncogene super-enhancers and maintain their high expression levels in tumor cells[15], explaining why BET inhibitors could specifically suppress tumor cell growth and induce apoptosis. Our recent study also demonstrates that BET inhibitors disrupt the Twist/BRD4 interaction and effectively inhibit invasion and cancer stem cell-like property of basal-like breast cancer (BLBC) cells[16]. Although it is

well accepted that BRD4-guided gene expression mediates diverse processes during tumor development and progression, whether and how BRD4 assembles transcriptional machinery on chromatin to activate feedback survival genes expression is totally unclear. Here our study discovered the novel role of FOXO3a/BRD4/CDK6 axis in AKTi resistance of luminal breast cancer cells.

## Results

**Bromodomain inhibitor enhances growth suppressive effects of AKTi.** As luminal subtype of breast cancer has activation of the PI3K/AKT pathway and the effect of monotherapy of PI3K/AKTis is moderate[17], we sought to identify additional strategies that may increase the efficacy of AKTi by rational combination therapies that will increase synergy and overcome emergence of resistance. We treated four luminal breast cancer cell lines (BT474, T47D, MDA-MB-453, and MDA-MB-361) with three different AKTi inhibitors (MK2206, AZD5363, and GSK690693). Consistent with previous studies that monotherapy of AKTi did not induce substantial growth suppression, about 50–80% of these tumor cells remain alive after treatment with these inhibitors (1 μM) for 4 days detected by both cell count and MTT (3-(4,5-dimethylthiazol-2-yl)-2,5-diphenyltetrazolium bromide) assays (Fig. 1a, b). Treatment with two individual BET inhibitors (JQ1 and MS417, 1 μM) also did not result in substantial suppression of growth and proliferation. However, the combination of JQ1 or MS417 with AKTi produces significant more growth suppressive effect than either single agent alone (Fig. 1a, b). These three AKTi have different mechanistic action on AKT inhibition. MK2206 suppresses the allosteric activation of AKT[18], AZD5363 binds to the catalytic pocket of AKT[19], and GSK690693 is an ATP-competitive pan-AKT kinase inhibitor, which exhibits efficacy irrespective of the mechanism of AKT activation involved[20]. Therefore, the synergistic effect seen in the combination of three different AKTi with BET inhibitors is not likely due to off-target effect. To further confirm this observation, we knocked down the expression BET family members BRD4, BRD3, BRD2, and BRDT individually and treated these cells with AKTi for 3 days. We found that only BRD4-knockdown greatly enhanced the suppressive effect of AKTi (Fig. 1c). These results indicate that BRD4 inhibition improves the growth suppressive effect mediated by AKTi, suggesting BRD4 and its related transcription machinery mediated oncogene expression is involved in AKTi resistance.

**Prolonged treatment of AKTi induces FOXO3a acetylation and its association with BRD4.** FOXO3a, a member of Forkhead family of transcription factors that regulates diverse gene expression in controlling various cellular processes, is a key target of AKT. AKT phosphorylates FOXO3 at three conserved serine/threonine residues (T32, S253, and S315), resulting in the binding of FOXO3 with 14-3-3 and thus the cytoplasmic retention as well as inactivation of this transcription factor[21,22]. Consistent with AKT inhibition, treatment with MK2206 (1 μM) in BT474 and T47D cells resulted in the nuclear translocation of FOXO3a in these cells (Fig. 2a). Our recent study identified that a histone H4 mimic GK-X-GK motif in Twist is responsible for the interaction of Twist with BRD4[16]. Intriguingly, we noticed that FOXO transcription factor family members also contain this GK-X-GK motif (Fig. 2b); notably, lysine 242 and 245, which are near the nuclear localization signal (NLS) of FOXO family proteins, have been identified as two major acetylation modification residues in FOXO3a protein previously and are involved in FOXO3a-mediated transcriptional activation[23–25], although the underlying mechanism remains unclear. We found that treatment with three different AKTi (1 μM) for 4 days in BT474 and T47D cells

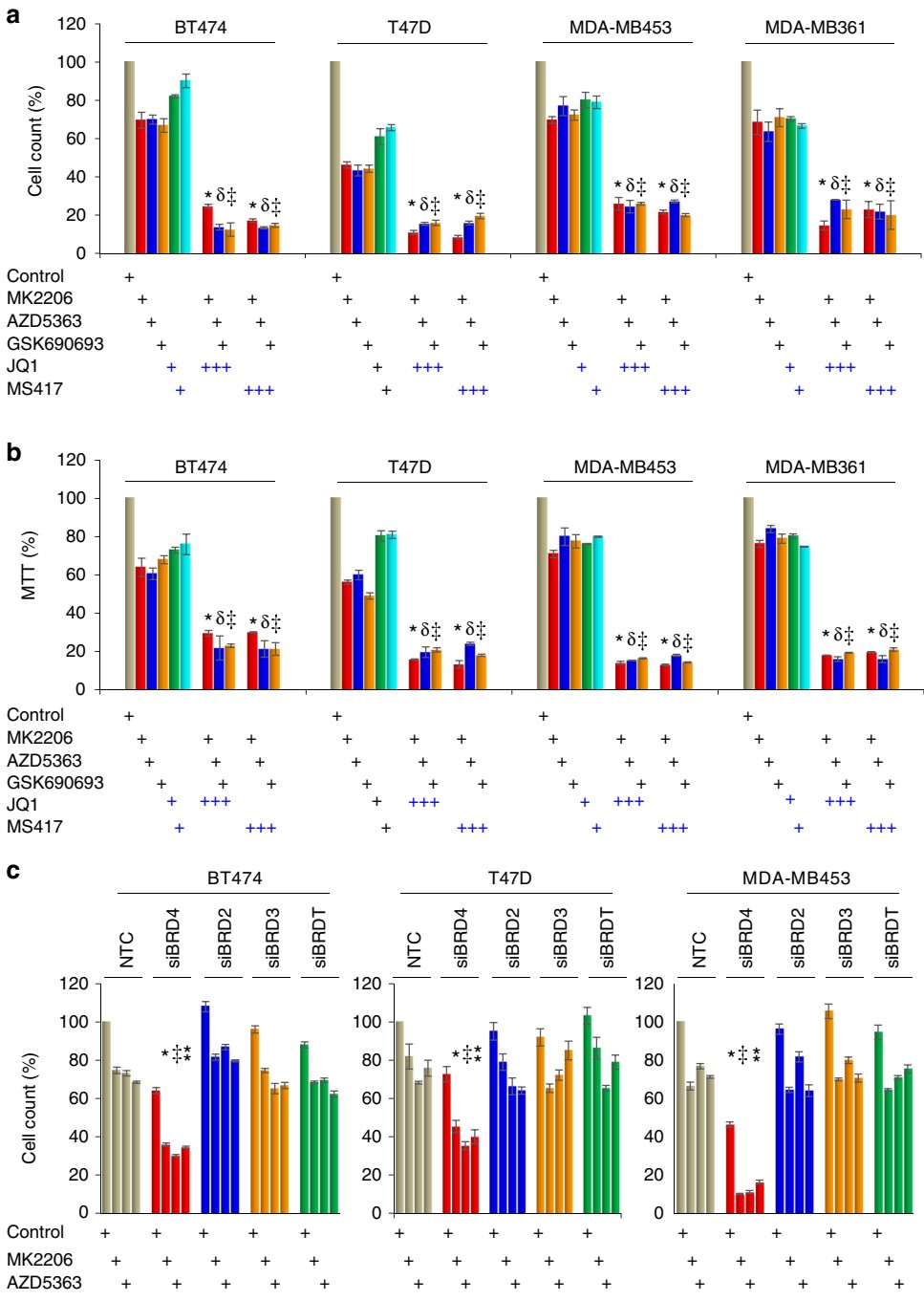

**Fig. 1** BET inhibitors increase the growth suppressive effects of AKT inhibitor (AKTi). **a**, **b** Breast cancer cell lines were treated with BET inhibitors (JQ1 or MS417, 1 μM) or AKTi (MK2206, AZD5363, and GSK690693, 1 μM), or in combination for 96 h. The growth suppressive effects were measured by either cell count (**a**) or MTT analyses (**b**). Data presented are representative of three experiments performed in quadruplicate as the mean ± SD. $^*/\square/\ddagger$ $p < 0.01$ when treatment group containing JQ1 or MS417 plus either MK2206 ($^*$), or AZD5363 ($\square$), or GSK690693 ($\ddagger$) is compared with the corresponding group with single inhibitor. **c** Expression of BRD4 family members was knocked down by siRNA in breast cancer cell lines, followed by treatment with various inhibitors (1 μM) for 96 h as described above. The growth suppressive effects were reflected by cell count assay. NTC stands for non-target control of siRNA (scramble). Data presented are representative of three experiments performed in triplicate as the mean ± SD. $^*/\ddagger/\ddagger$ $p < 0.01$ when group with BRD4-knockdown plus either MK2206 ($^*$), or AZD5363 ($\ddagger$), or GSK690693 ($\ddagger$) is compared with other knockdown groups in combination with same inhibitor

significantly induced FOXO3a acetylation and BRD4 association, concomitant with the suppression of FOXO3a phosphorylation (Fig. 2c). The level of endogenous FOXO3a acetylation and its interaction with BRD4 induced by AKTi appeared at day 1 and gradually reached to maximum at day 4 after MK2206 treatment

(Fig. 2d). This finding suggests that the FOXO3a-BRD4 interaction became eminent after prolonged exposure to AKTi, and that this interaction may be involved in drug resistance against AKTi. The FOXO3a-BRD4 interaction is sensitive to bromodomain blockade, because the addition of JQ1 or MS417 (2 μM) in the

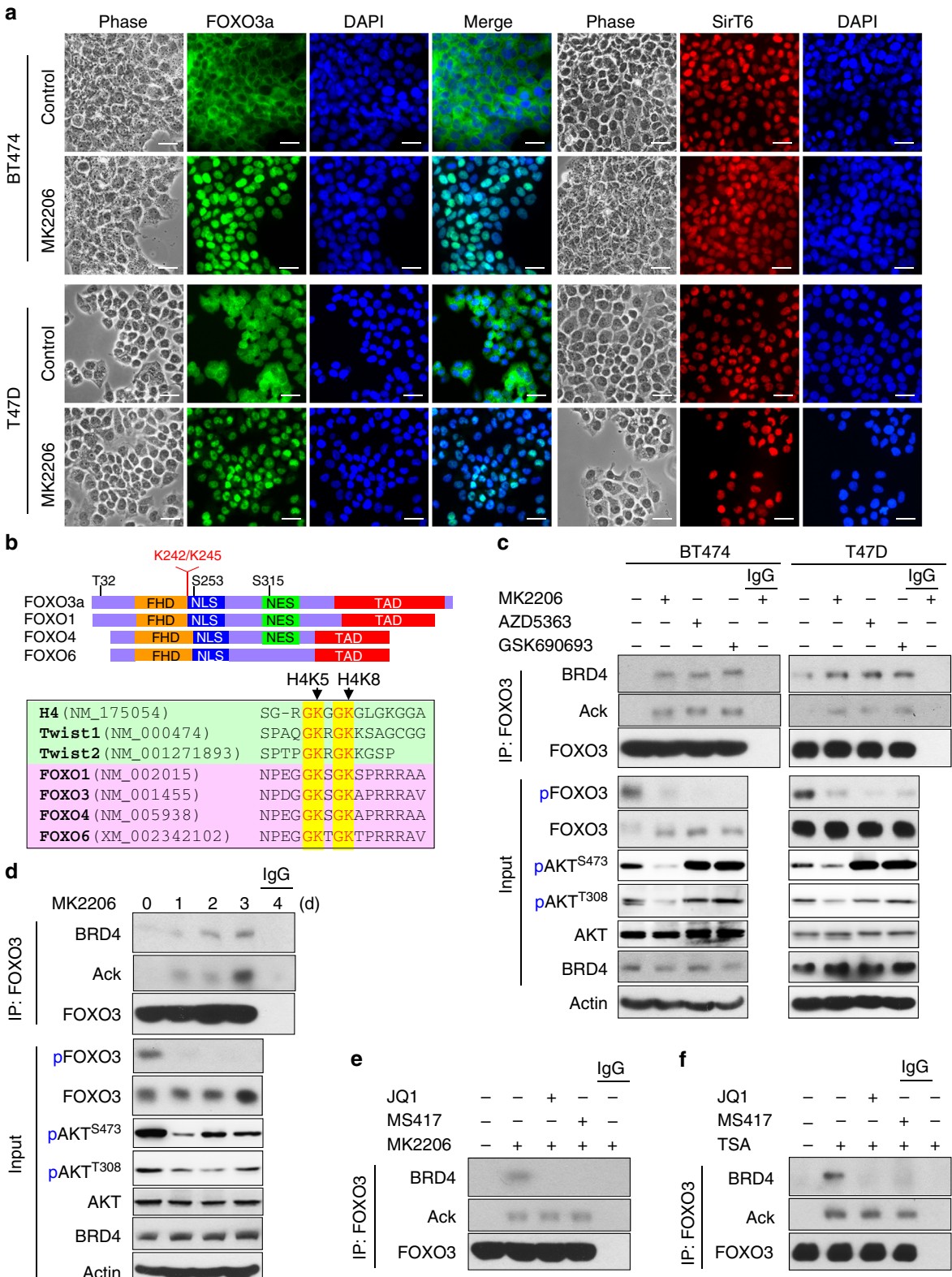

immunoprecipitation (IP) buffer completely disrupts the FOXO3a-BRD4 interaction in BT474 cells (Fig. 2e and Supplementary Figure 1). To further extend this observation, we treated cells with de-acetylase inhibitors Trichostatin A (TSA) (2 μM) and nicotinamide (4 mM) overnight. Endogenous FOXO3a became robustly acetylated and its interaction with BRD4 was disrupted by adding JQ1 or MS417 in the IP buffer (Fig. 2f and Supplementary Figure 1). These data indicate that AKTi treatment induces endogenous FOXO3a nuclear translocation, acetylation, and its interaction with BRD4, which can be disrupted by BET inhibitors.

**Fig. 2** Prolonged treatment with AKTi induces FOXO3a acetylation and BRD4 association. **a** BT474 and T47D cells were treated with 1 μM MK2206 for 6 h, the cellular localization of FOXO3a (green) and SirT6 (red) were analyzed by immunofluorescent staining. Nuclei stained with DAPI (blue). Scale bar, 50 μM. **b** Schematic diagram showing the domain structure of FOXO family members. Sequence alignment indicates the highly conserved GK-X-GK motif among K5/8 of histone H4, K73/76 of Twist, K242/245 of FOXO3a, in which the lysine residues have been reported to be acetylated in vivo. **c** BT474 and T47D cells were treated with AKTi for 4 day, cell extracts were subjected to immunoprecipitation (IP) with FOXO3a antibody, the acetylation of endogenous FOXO3a and its interaction with BRD4 were analyzed by western blot using pan acetylated-lysine and BRD4 antibodies. **d** BT474 cells were treated with 1 μM MK2206 for various time intervals. Cell extracts were prepared for IP with FOXO3a antibody, the acetylation of endogenous FOXO3a and bound BRD4 were analyzed by western blotting. **e** BT474 cells were treated with 1 μM MK2206 for 4 days. Cell extracts were prepared for IP with FOXO3a antibody in a buffer containing either 2 μM JQ1 or MS417. Western blottings were used to detect FOXO3a acetylation and its interaction with BRD4. **f** BT474 cells were treated with 2 μM of TSA and 4 mM of Nicotinamide overnight. Cell extracts were prepared and FOXO3a were IP in a buffer containing either JQ1 or MS417 as described above. FOXO3a acetylation and its interaction with BRD4 were analyzed by western blotting

**BD2 of BRD4 recognizes K242/245 di-acetylated FOXO3a.** BRD4 contains two tandem bromodomains (BD1 and BD2) that are responsible for interaction with di-acetylated substrates. To determine which bromodomain of BRD4 is responsible for the FOXO3a-BRD4 interaction, we co-expressed FOXO3a with either BD1 or BD2 of BRD4 in HEK293 cells, and found that FOXO3a interacted with BD2 but not BD1 of BRD4, and the interaction could be disrupted by the addition of JQ1 or MS417 in the IP buffer (Fig. 3a). As lysine 242 and 245 (K242/245) of FOXO3a are located in GK-X-GK motif similar to Twist K73/76 (Fig. 2b), we co-expressed wild-type (WT) or KR mutant (in which K242/245 were changed to arginine) of FOXO3a with BRD4 in HEK293 cells. We found that MK2206 treatment induced the acetylation of WT-FOXO3a and its interaction with BRD4; however, the acetylation of KR mutant and its interaction with BRD4 were completed abolished (Fig. 3b). In addition, treatment of TSA and nicotinamide, which inhibits the deacetylation event in cells, induced robust acetylation of WT-FOXO3 and its interaction with BD2 of BRD4 (Fig. 3c). This effect was completely abolished in KR-FOXO3a, indicating that K242/245 di-acetylation of FOXO3a is critical for the interaction with BD2 of BRD4.

We also performed a two-dimensional $^1$H-$^{15}$N heteronuclear single quantum coherence nuclear magnetic resonance (NMR) titration study and showed that BRD4-BD2 exhibited substantially more extended chemical shift perturbations upon binding to the di-acetylated FOXO peptides than those produced by BRD4-BD1, including K242/245 of FOXO3a (Fig. 3d). Our analyses indicated that the three-dimensional (3D) NMR structure of the BRD4-BD2 bound to a FOXO3a-K242ac/K245ac peptide and confirmed that indeed the BRD4-BD2 recognition of the di-acetylated site of FOXO3a is highly similar to that of Twist. The key residues of His437 and Asn433 at the acetyl-lysine binding site that are responsible for the BRD4-BD2's specificity for this molecular recognition are colored in red and green, respectively (Fig. 3e and Supplementary Figure 2). To further confirm these observations, we generated a His437 and Asn433 double-mutant of full-length BRD4 (HN), in which both His437 and Asn433 were changed to Alanine. Compared with WT-BRD4, HN-BRD4 completely lost the interaction with FOXO3a in a co-IP experiment (Fig. 3f). These results indicate that the FOXO3a-BRD4 interaction is dependent on BD2 of BRD4-mediated recognition of di-acetylated K242/245 of FOXO3a. A recent study applied motif analyses and predicted the involvement of FOXO transcription factors in BRD4-enriched regions on chromatin[26]. Our study elucidates the exact molecular basis of combined action of FOXO3a and BRD4.

**AKTi induces FOXO3a acetylation by disrupting the FOXO3a-SirT6 interaction.** To define the underlying mechanism responsible for AKTi-induced FOXO3a acetylation, we co-expressed FOXO3a with seven SirT de-acetylase family members in HEK293 cells, we found only SirT6 and SirT7 (to a less

extent) associate with FOXO3a (Fig. 4a). With IP endogenous FOXO3a from BT474 and T47D cells, only endogenous SirT6 but not SirT7 was detected (Fig. 4b and data not shown). Interestingly, all three AKTi (1 μM) blocked the endogenous interaction of FOXO3a with SirT6. To examine whether SirT6 is a de-acetylase of FOXO3a, we knocked down endogenous SirT6 in BT474 cells and found MK2206-induced FOXO3a acetylation was greatly elevated (Fig. 4c). In an in vitro deacetylation assay, the addition of purified active SirT6 protein removed the acetylation of endogenous FOXO3a protein immunoprecipitated from BT474 cells induced by TSA and nicotinamide treatment (Fig. 4d). These data indicate that SirT6 interacts with and de-acetylates FOXO3a, which inhibited by AKTi.

When AKT is inhibited, FOXO3a phosphorylation can be quickly removed by PP2A, resulting in the disassociation of 14-3-3, the exposure of NLS of FOXO3a and the consequent nuclear translocation of this molecule[27]. To further investigate the FOXO3a-SirT6 interaction, we expressed either WT or TM mutant of FOXO3a (T32/S253/S315 were mutated to alanine, which detains FOXO3a in the nucleus) in cells. WT-FOXO3a is mainly localized in the cytoplasm, whereas TM-FOXO3a is exclusively resided in the nuclear (Supplementary Figure 3). AKT inhibition induces nuclear translocation of WT-FOXO3a. However, endogenous SirT6 exclusively resides in the nucleus even in the presence of AKTi (Fig. 2a and Supplementary Figure 3). Interestingly, when WT- and TM-FOXO3a are immunoprecipitated, TM mutant binds to SirT6 much stronger than WT-FOXO3a (Fig. 4e). MK2206 treatment disrupted the interaction of SirT6 with both WT and TM mutant of FOXO3a, suggesting that the disruption of FOXO3a-SirT6 interaction by MK2206 is independent of FOXO3a phosphorylation and cellular localization. A recent study indicated that S338 of SirT6 is phosphorylated by AKT, and that this phosphorylation is critical for SirT6 to interact with its binding partners[28,29]. In line with this finding, we found that IGF-1 treatment in BT474 and T47D cells induces SirT6-S338 phosphorylation detected by antibodies against S338 phosphorylation and AKT phospho-substrates; this phosphorylation was abolished by pre-treatment with MK2206 (1 μM) (Fig. 4f). In addition, the endogenous interaction of AKT with SirT6 was completely disrupted by MK2206 in BT474 and T47D cells (Fig. 4g). To further extend this observation, we co-expressed WT, S338A (cannot be phosphorylated), and S338E (mimic phosphorylation) SirT6 with FOXO3a in HEK293 cells. WT-SirT6 interacted with FOXO3a and this interaction was reduced by MK2206. However, S338A lost this interaction whereas S338E remained bound to FOXO3a partially even in the presence of MK2206 (Fig. 4h). Together, these data suggest an interesting model, in which under normal condition, FOXO3a diffuses in the whole cells, part of them associates with SirT6. Upon extracellular stimuli, hyper-activated AKT phosphorylates and detains FOXO3a in the cytosol; meanwhile, it maintains the phosphorylation status and substrate-binding activity of SirT6. On the

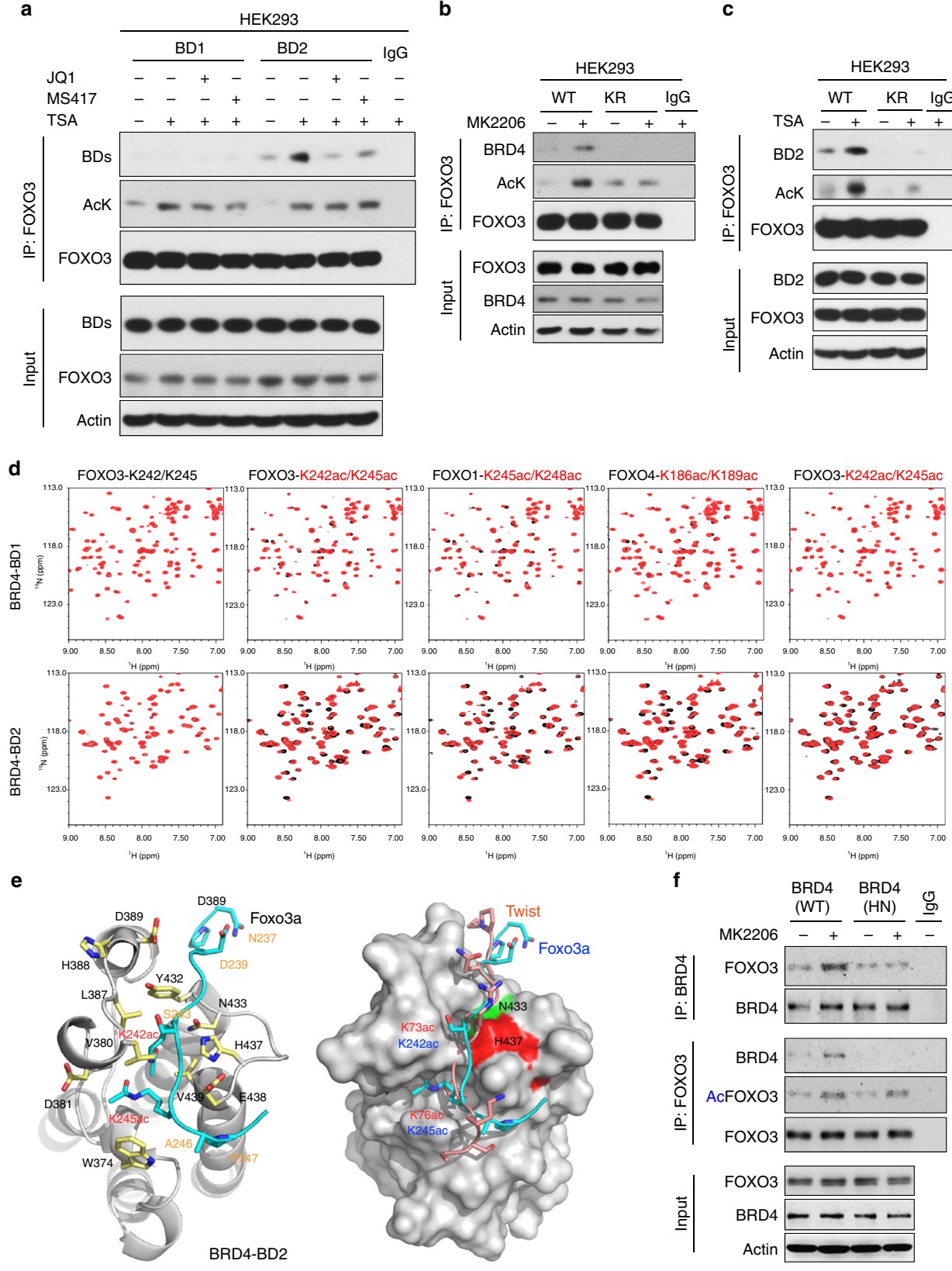

contrary, AKT inhibition enhances FOXO3a nuclear translocation, but it also reduces the ability of SirT6 targeting to its substrate, resulting in the separation of SirT6 from FOXO3a, although they both reside in the nucleus, ultimately leading to the increased FOXO3a acetylation and its interaction with BRD4 (left panel, Fig. 5a). These observations elucidate the exquisite interplay among AKT, SirT6, and FOXO3a, renovate the understanding on regulation of activity of FOXOs protein, implicating the potential mechanism of drug resistance to AKT inhibition.

**Fig. 3** BD2 of BRD4 recognizes and interacts with K242/245 di-acetylated FOXO3a. **a** HA-tagged FOXO3a was co-expressed with Flag-tagged BD1 and BD2 in HEK293 cells treated with 2 μM TSA and 4 mM Nicotinamide overnight. After IP with anti-HA antibody, FOXO3a acetylation and its association with BD1 and BD2 were examined by western blotting using pan acetylated-lysine and anti-Flag antibodies. **b** Wild-type (WT) or K242R/245 R mutant (KR) of FOXO3a were co-expressed with Flag-tagged BRD4 in HEK293 cells treated with 1 μM MK2206 for 2 day. After IP with anti-HA antibody, FOXO3a acetylation and its association with BRD4 were examined by western blotting using pan acetylated-lysine and anti-Flag antibodies. **c** WT or KR mutant of FOXO3a was co-expressed with Flag-tagged BD2 in HEK293 cells treated with 2 μM TSA and 4 mM Nicotinamide overnight. After IP with anti-HA antibody, FOXO3a acetylation and its association with BD2 were examined by western blotting. **d** The FOXO3a-binding selectivity of the BD1 and BD2 of BRD4. 2D 1H-15N HSQC spectra of BRD4-BD1 or BD2 in the free form (black) and in the presence of a FOXO3a peptide (residues 238–249, NPDGG-K242ac-RG-K245ac-APR) that is non-acetylated, or contains single or di-acetylated K242 or K245 with protein:ligand molar ratio of 1:5 (red). Other FOXO members were also included for analyses. **e** Left and middle panel: Stereo ribbon diagram of the 3D solution structure of the BRD4-BD2 bound to a di-acetylated K242ac/K245ac FOXO3a peptide (cyan). Side chains of key residues engaged at the protein/peptide interactions are depicted and color-coded by atom type. Right panel: surface-filled representation of the 3D structure of the BRD4-BD2 highlights its same mode of ligand recognition for FOXO3a (cyan) and Twist (yellow) peptides. **f** FOXO3a was co-expressed with full-length WT- or HN-BRD4 in HEK293 cells treated with 1 μM MK2206 for 2 day. After FOXO3a and BRD4 were immunoprecipitated, the associated BRD4 and FOXO3a were examined by western blotting

**AKTi induces CDK6 expression**. To test whether the interaction of SIRT6 with FOXO3a is involved in the drug resistance of AKTi, we expressed vector, WT-SirT6, S338A-SirT6, or S338E-SirT6 in BT474 and T47D cells then treated them with MK2206 (1 μM) for 4 days. Expression of WT-SirT6, S338A-SirT6, or S338E-SirT6 did not affect the overall cell growth in the absence of MK2206 compared with vector control (middle and right panel, Fig. 5a). Although WT-SirT6 expression conferred partial sensitivity to MK2206-mediated growth inhibition, expression of S338E-SirT6 endowed cells with more sensitivity to MK2206, whereas S338A-SirT6 expressing cells became completely insensitive to MK2206. These results support the notion that S338 phosphorylation of SirT6 increases its interaction with and deacetylation of FOXO3a, which prevents BRD4 interaction and suppresses the transcriptional activity of FOXO3a.

To further investigate the mechanism of FOXO3a-BRD4 interaction in mediating AKTi resistance, we sought to identify the target genes of the FOXO3a-BRD4 complex by performing RNA-sequencing analyses on BT474 and T47D cells treated with MK2206 for 3 days (GSE118148) (Fig. 5b and Supplementary Figure 4). We reasoned that survival or growth-promoting genes that are transcriptionally induced by FOXO3a could have a role in AKTi resistance. Among 78 overlapping genes that differentially expressed in more than 2.5-fold in both cell lines, CDK6 is noted as an oncogenic kinase that governs G1/S phase transition and cell cycle progression[9,10]. Consistent with RNA-sequencing analysis, MK2206 (1 μM) treatment for 3 days induced both mRNA and protein levels of CDK6 in BT474, T47D, and ZR75 cells (Fig. 5c). Although CDK6 is found to be upregulated in BLBC compared with luminal subtype of breast cancer[30], we did not observed CDK6 induction in two BLBC cell lines (MDA-MB231 and BT549) treated with MK2206 for up to 4 days (Supplementary Figure 5a). This suggests that MK2206-mediated CDK6 upregulation is specific to luminal subtype of breast cancer. The induction of CDK6 by MK2206 was time dependent, which began at day 1 and reached maximal at day 4 in BT474 cells (Fig. 5d). Interestingly, the pattern of CDK6 induction is similar to the increased FOXO3a acetylation and its interaction with BRD4 mediated by MK2206 (Fig. 2d), suggesting that CDK6 expression is regulated by the FOXO3a-BRD4 complex. Consistent with CDK6 induction, E2F driven luciferase reporter activity was upregulated in cells treated with MK2206 (Fig. 5e) and the increased E2F-driven luciferase activity can be inhibited by either JQ1 or CDK6 inhibitor Palbociclib. These results suggest that the FOXO3a-BRD4 interaction is critical for the expression of CDK6, which can phosphorylate Rb and result in E2F activation. Indeed, treatment with JQ1 or MS417 almost completely blocked MK2206-mediated CDK6 induction in

BT474 and T47D cells (Fig. 5f). The CDK6 induction is not only limited to MK2206, because two other AKTi (1 μM) (AZD5363 and GSK690693) also induced CDK6 expression, which could be inhibited by either JQ1 or MS417 (Fig. 5g). Together, these findings indicate that AKTi-induced CDK6 expression is likely transcriptionally activated by the FOXO3a-BRD4 complex.

**CDK6 is a targeted gene of the FOXO3a-BRD4 complex**. We noticed two typical FOXO3a-binding sites in the CDK6 gene promoter (Fig. 6a). When CDK6 promoter luciferase reporter was co-expressed with BRD4 and WT or KR mutant of FOXO3a in HEK293 cells, FOXO3a and BRD4 barely enhanced CDK6 reporter activity in the absence of MK2206. However, MK2206 significantly induced CDK6 promoter luciferase activity in the presence of FOXO3a; the addition of BRD4 further increased the luciferase activity (Fig. 6b). KR-FOXO3a, which cannot interact with BRD4, lost its ability to induce CDK6 promoter luciferase activity. We also constructed three mutant CDK6 promoter luciferase reporters (M1, M2, and M3), in which the first and the second FOXO3a-binding element were mutated, respectively, or both (Fig. 6a). Luciferase assay indicated that M1 and M3 totally lost reporter activity, whereas M2 retained large percent of activity, suggesting that the FOXO3a-binding element distal to the transcription starting site is the major binding site for the FOXO3a-BRD4 complex at the CDK6 promoter (Fig. 6c). Indeed, chromatin IP from BT474 and T47D cells displayed that FOXO3a, BRD4, and RNA polymerase II are targeted to this region of CDK6 promoter upon MK2206 treatment, and that these associations were abolished by JQ1 (Fig. 6d). We also performed the sequential chromatin immunoprecipitation (ChIP) analyses to further validate the association of the FOXO3-BRD4 complex at the CDK6 promoter (Fig. 6e). Using a control IgG in the first cycle of ChIP, we did not detect the association of FOXO3a or BRD4 at the CDK6 promoter in the second cycle of ChIP. However, using either FOXO3a or BRD4 antibody in the first cycle of ChIP, we found the interaction of BRD4 or FOXO3a, respectively, at the CDK6 promoter in the sequential ChIP. In addition, we knocked down FOXO3 in BT474 cells and found that FOXO3-knockdown significantly reduced BRD4 association at the CDK6 promoter (Fig. 6f). Together, these data further strengthen our findings that the interaction of FOXO3a-BRD4 complex is critical for the transcription of CDK6.

To further validate the transcriptional activation of CDK6 by the FOXO3a-BRD4 complex, we knocked down FOXO3a in BT474 cells, and then followed up with a rescue expression of WT- or KR-FOXO3a in these cells. FOXO3a knockdown inhibited MK2206-induced CDK6 expression (Fig. 6g); rescue

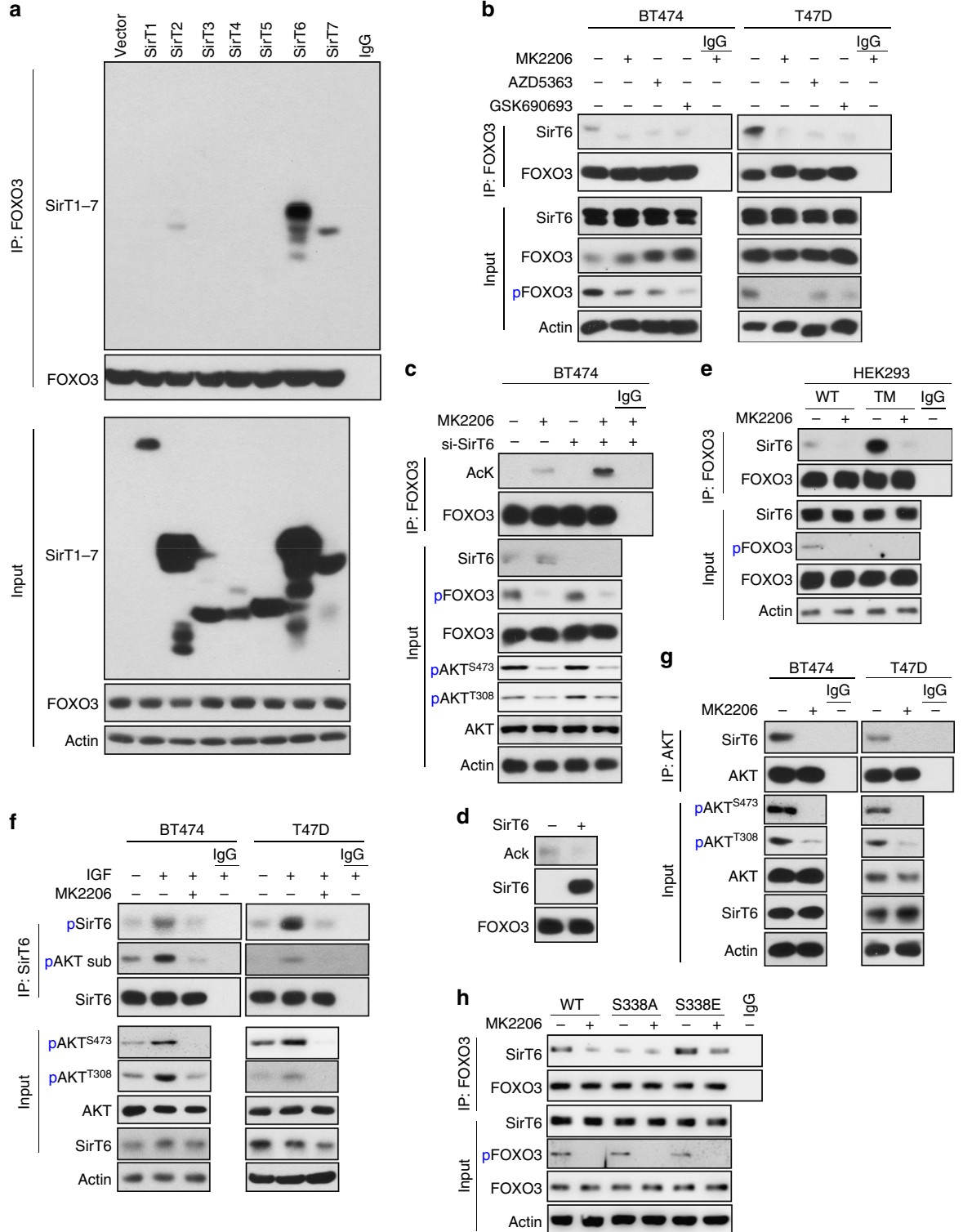

expression of WT- but not KR-FOXO3a, completely restored CDK6 expression. In addition, FOXO3a knockdown enhanced the sensitivity of cells to MK2206-induced growth inhibition, re-expression of KR mutant lost the resistance ability as WT-FOXO3a did (bottom panel, Fig. 6g). We also knocked down BRD4 expression in BT474 and T47D cells (Supplementary Figure 5b and Fig. 6h). BRD4 knockdown completely abolished MK2206-induced CDK6 expression. Rescue expression of WT-BRD4 but not HN-BRD4, which cannot interact with FOXO3a,

restored CDK6 expression in BT474 cells (Fig. 6h). Together, these results indicate that AKTi-induced CDK6 expression is mediated through the association of the FOXO3a-BRD4 complex at the CDK6 promoter and disruption of the FOXO3a-BRD4 complex by JQ1 or MS417 abolishes CDK6 induction.

**Inhibition of CDK6 confers drug sensitivity to AKTi.** To further determine the biological role of CDK6 induction in

**Fig. 4** AKTi induces FOXO3a acetylation by disrupting the SirT6/FOXO3a interaction. **a** Flag-tagged Sirt family members were co-expressed with HA-FOXO3a in HEK293 cells. After IP with HA antibody, the bound Sirt members were examined by western blotting using anti-Flag antibody. **b** After BT474 and T47D cells were treated with various AKTi for 4 day, cell extracts were subjected to IP with FOXO3a antibody, binding of endogenous SirT6 was analyzed by western blotting. **c** Expression of endogenous SirT6 was knocked down by siRNA in BT474 cells, followed by treatment with 1 μM MK2206 for 2 days. After IP with FOXO3a antibody, endogenous FOXO3a acetylation was analyzed by western blotting using pan acetylated-lysine antibody. **d** Acetylated FOXO3a was purified from TSA and Nicotinamide-treated BT474 cells by FOXO3a antibody and then incubated with purified recombinant SirT6 protein in a deacetylation assay as described in the Experimental Procedures. The acetylated state of FOXO3a was assessed by western blotting using pan acetylated-lysine antibody. The immuno-purified FOXO3a and SirT6 used in this assay were analyzed by western blotting. **e** WT or TM mutant (with T32, S253, and S315 of FOXO3a changed to alanine, which confers FOXO3a localized exclusive in the nucleus) of FOXO3a were expressed in HEK293 cells followed by treatment with 1 μM MK2206. After IP FOXO3a, the association of SirT6 was analyzed by western blotting using anti-Flag antibody. **f** BT474 and T47D cells were pre-treated with 1 μM MK2206 followed by stimulation with 10 ng/ml of IGF-1 for 4 h, cell extracts were subjected to IP with SirT6 antibody, phosphorylation of endogenous SirT6 was analyzed by either specific SirT6-S338 phosphorylation or phospho-AKT substrate antibody by western blotting, respectively. **g** BT474 and T47D cells were treated with 1 μM MK2206, cell extracts were subjected to IP with AKT antibody, the bound SirT6 was analyzed by western blotting. **h** WT, S338A, or S338E mutant of SirT6 were expressed in HEK293 cells followed by treatment with or without 1 μM MK2206. After IP FOXO3a, the bound SirT6 was analyzed by western blotting using anti-Flag antibody

AKTi-mediated drug resistance, we generated MK2206-resistant cells in T47D and ZR75 cell lines by exposing them with gradually increasing concentration of MK2206 (from 1 μM to 20 μM) over a 4-month period (left panel, Fig. 7a). Both resistant cell lines can grow in the presence of 2 μM MK2206 with no apparent growth inhibition, because with 2 μM MK2206 treatment, significantly fewer of these cells have decreased S-phase of cell cycle compared with that of the parental cells (middle panel, Fig. 7a). This is likely due to the induction of CDK6, which phosphorylates Rb (serine 780) and disrupts cell cycle arrest (left panel, Fig. 7a). Intriguingly, JQ1 or Palbociclib conferred sensitivity of these MK2206-resistant cells (right panel, Fig. 7a). To validate this finding, we generated CDK6-expression stable clones in BT474 and T47D cells (left panel, Fig. 7b). Again, these CDK6-expressing cell lines induced Rb phosphorylation and have a lower population of cells with decreased S-phase when exposed to 2 μM MK2206 (middle panel, Fig. 7b). As expected, CDK6-expressing cell lines were less sensitive to MK2206 and they were also less sensitive to growth inhibition mediated by the combination of MK2206 and JQ1 (right panel, Fig. 7b). This is because exogenous CDK6 expression was not under the control of endogenous CDK6 promoter, which can be activated by the FOXO3a-BRD4 complex upon AKTi. However, these CDK6-expressing cell lines remained highly sensitive to the growth suppression mediated by the combination of MK2206 and Palbociclib. These results clearly indicate that AKTi-induced CDK6 expression, which is mediated by the FOXO3a-BRD4 complex, is critical for drug resistance to AKT inhibition.

To further extend our finding that the FOXO3a-BRD4-CDK6 axis promotes resistance of PI3K/AKT inhibition, we treated BT474 cells with Lapatinib (1 μM), which is an EGFR/HER2 dual tyrosine kinase inhibitor that is commonly used in breast cancer treatment. Similar to AKTi, prolonged treatment of Lapatinib for 4 days induced FOXO3a acetylation and its interaction with BRD4, as well as the consequent CDK6 induction (Fig. 7c). Combination of JQ1 or Palbociclib greatly enhanced the growth suppressive effect mediated by Lapatinib. Together, our data not only confirm that CDK6 is a major feedback survival factor against AKTi, but also suggest that the FOXO3a-BRD4-CDK6 axis is a common drug resistance mechanism of PI3K/AKT targeting.

**Pharmacological inhibition of BRD4/FOXO3a or CDK6 sensitizes luminal breast cancer cells to AKTi in vivo.** To extend our finding in vivo, we inoculated BT474 cells into the 6-week-old female athymic mice subcutaneously. When tumors reached 250 mm³,

the mice were equally divided into six groups (six mice/group) and received the following treatments: (1) control vehicle; (2) JQ1 (50 mg/kg); (3) Palbociclib (100 mg/kg); (4) MK2206 (90 mg/kg); (5) MK2206 plus JQ1; and (6) MK2206 plus Palbociclib for 2 weeks. Consistent with the cell culture studies in vitro, mono-therapy with JQ1, Palbociclib, or MK2206 slightly reduced tumor growth; however, the combinatory treatments of, either MK2206 plus JQ1 or MK2206 plus Palbociclib, significantly inhibited tumor growth (Fig. 8a, c). To further explore the clinical relevance of CDK6 expression, we performed immunohistochemistry (IHC) analysis of a tissue microarray featuring of 343 cases of breast tumor samples that contains 56 cases of recurrence disease from Kentucky. We scored different levels of CDK6 expression in these tumor samples and found that high CDK6 expression correlated with recurrent disease in patients with ER + /PR + /HER − luminal subtype of breast cancer (Fig. 8d, e). These in vivo data support our findings in cell culture experiments and further strengthen the hypotheses that the FOXO3a-BRD4-CDK6 axis is critical for conferring drug resistance to AKTi and inhibitors of FOXO3a-BRD4 and CDK6 can sensitize the growth suppressive effects mediated by AKTi in luminal breast cancer.

## Discussion
Our study reveals a novel mechanistic insight for drug resistance mediated by the FOXO3a-BRD4-CDK6 signaling axis. In proliferating tumor cells, AKT not only phosphorylates FOXO3a to retain it in the cytoplasm, but also phosphorylates SirT6 at serine 338 to maintain its substrate-binding ability. When cells are treated with AKTi, FOXO3a undergoes de-phosphorylation and translocates into the nucleus; at the same time, de-phosphorylated SirT6 loses its capacity to associate with FOXO3a, resulting in the increased FOXO3a acetylation. Acetylated FOXO3a interacts with BD2 of BRD4, forming a transcription activation complex with pTEFb and RNA polymerase II on the CDK6 gene promoter and activates CDK6 transcription (Fig. 8f).

Our study provides several potential therapeutic strategies for overcoming drug resistance in luminal breast cancer. Luminal breast tumors have been shown to be dependent on the PI3K-AKT-mTOR signaling for survival and proliferation; therefore, they are highly sensitive to inhibition of pathway components. Although corresponding small inhibitors have been shown to possess a robust efficiency in killing tumor cells, intrinsic drug resistance, a hallmark associated with poor clinical outcomes, inevitably arises. We found that individual treatment with BET or CDK6 inhibitors have only a mild suppressive effect on proliferation and survival of luminal breast cancer. However, either

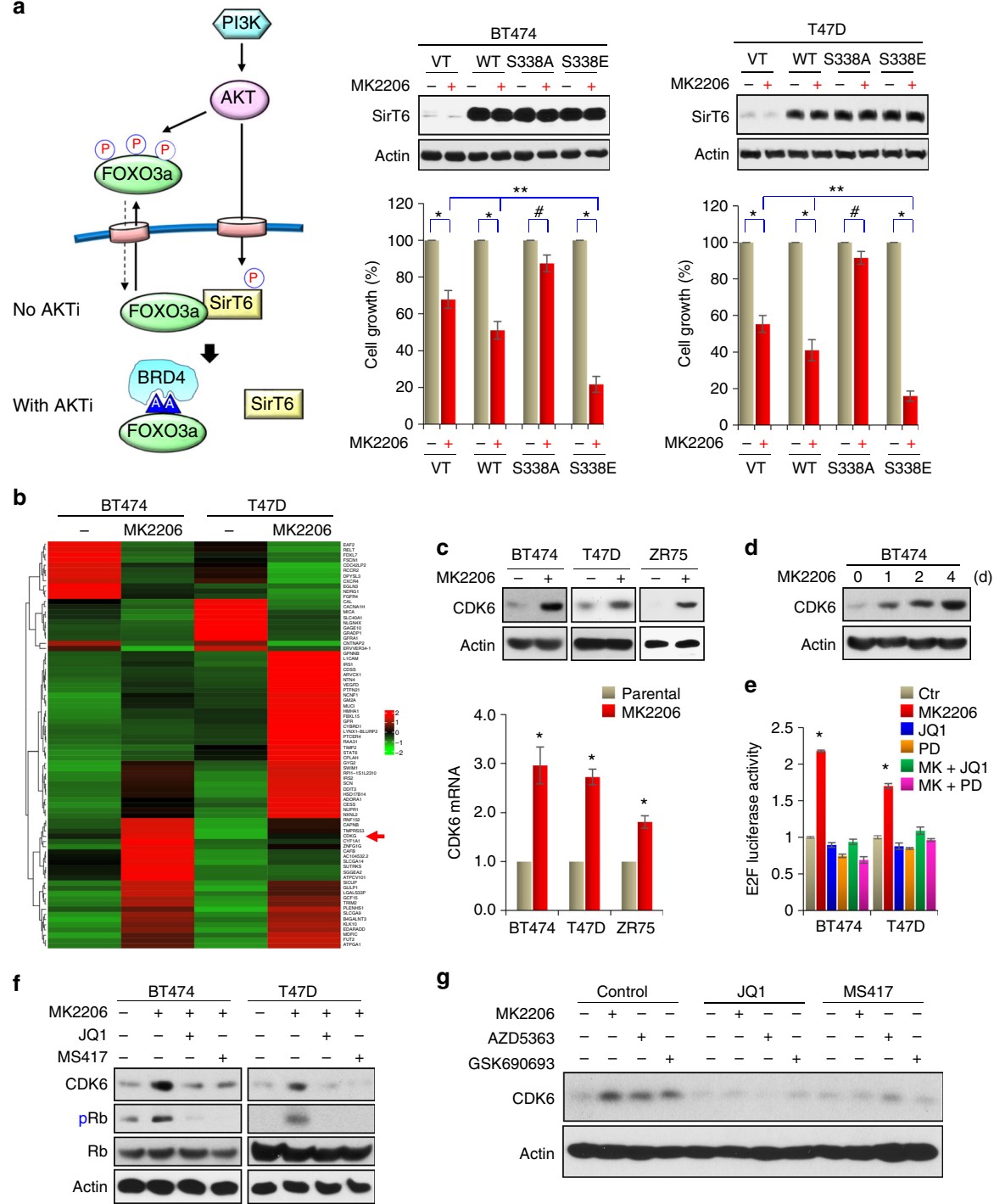

compound exhibits a synergistic effect with AKTi in the inhibition of cell growth in vitro and in vivo, indicating that the induction of CDK6 by the FOXO3a-BRD4 complex has a critical role in the development of intrinsic AKTi resistance. Targeting either the FOXO3a-BRD4 complex or CDK6 greatly potentiates the anti-proliferative and pro-apoptotic effects of AKTi.

Our study also provides new insight into the pro-survival role of FOXO3a for tumor cells. FOXO proteins have long been considered tumor suppressors, because they inhibit tumor cell proliferation and induce apoptosis, which are negatively regulated by AKT, a central regulator of cell proliferation and survival.

Specifically, FOXOs can inhibit proliferation by inducing expression of cell cycle inhibitors p15, p19, p21, and p27, while repressing cyclin D. FOXOs also induces apoptosis by upregulating expression of several pro-apoptotic molecules, including FasL, TRAIL, Puma, and Bim[31]. Unexpectedly, we identified a pro-survival function of FOXO3a in tumor cells via an upregulation of *CDK6* gene expression. CDK4/6 function as the major oncogenic drivers in cancer cells by promoting G1-S cell cycle transition and by preventing cancer cell-intrinsic tumor suppression. For example, CDK4/6 prevent senescence and apoptosis by activating downstream signaling such as Rb/E2F and

**Fig. 5** AKTi induces CDK6 expression. **a** A proposed model illustrating the interplay of FOXO3a-AKT-SirT6, which leads to the formation of the FOXO3a-BRD4 complex (left panel). WT, S338A, or S338E mutant of SirT6 were expressed in indicated cells followed by treatment with 1 μM MK2206 for 4 days. The growth suppressive effects were examined by cell count analyses. SirT6 expression was confirmed by western blotting. Data presented are representative of three experiments performed in triplicate as the mean ± SD. $^*p < 0.01$ (MK2206-treated group is compared with control group without treatment); $^\#p > 0.05$ (MK2206-treated S338A-SirT6 group is compared with control group without treatment); and $^{**}p < 0.01$ (MK2206-treated S338E-SirT6 group is compared with MK2206-treated vector and WT-SirT6 groups). **b** BT474 and T47D cells were treated with 1 μM MK2206 for 3 days, differentially expressed genes with more than 2.5-fold changes on RNA-sequencing analysis (GSE118148) from both cell lines were used to identify potential FOXO3a target genes shown in the heat-map. **c** BT474, T47D, and ZR75 cells were treated with 1 μM MK2206 for 3 days. The mRNA and protein levels of CDK6 were analyzed by RT-PCR and western blotting, respectively. Data presented are representative of three experiments performed in triplicate as the mean ± SD. $^*p < 0.01$ when MK2206-treated group is compared with control group. **d** BT474 cells were treated with 1 μM MK2206 for different time intervals. CDK6 expression was analyzed by western blotting. **e** E2F driven luciferase reporter was expressed in indicated cells treated with MK2206, JQ1, Palbociclib (PD), or in combination. Luciferase activities were determined by dual luciferase assay. Data presented are representative of three experiments performed in triplicate as the mean ± SD. $^*p < 0.01$ when MK2206-treated group is compared with other groups. **f** BT474 and T47D cells were treated with MK2206 (1 μM) in the absence or presence of JQ1 or MS417 (1 μM). CDK6 expression was analyzed by western blotting. **g** BT474 cells were treated with three different AKTi (1 μM) in the absence or presence of JQ1 or MS417 (1 μM). CDK6 expression was analyzed by western blotting

FOXM1[32]. Numerous studies have shown that CDK4/6 are aberrantly active in many different forms of human cancers, largely due to high levels of cyclin D, an important cofactor for CDK4/6[33]. Although it is clear that the increase in cyclin D in tumor cells is regulated at transcriptional, translational, and posttranslational levels[32], regulation of CDK4/6 at the transcriptional level remains unclear. Our data show that the *CDK6* gene promoter contains specific FOXO3a-binding sites. BRD4 and FOXO3a cooperatively target the CDK6 promoter, forming a transcriptional activation complex to induce *CDK6* gene transcription. This process consequently leads to increased Rb phosphorylation and elevated E2F transcriptional activity. The CDK6 induction by FOXO3a/BRD4 is responsible for tumor cell survival because overexpression of CDK6 in luminal breast cancer cells significantly relieves AKTi-induced cell growth arrest and apoptosis. CDK6 inhibition greatly potentiates the anti-cancer effects of AKTi. In addition, CDK6 expression is robustly increased in MK2206-resistant cell lines. Palbociclib significantly restores the sensitivity of tumor cells to AKTi. Furthermore, high CDK6 protein level is tightly correlated to the recurrence of luminal breast cancer, lending further support to the pro-survival role of CDK6 mediated by the FOXO3a-BRD4 complex in luminal breast cancer.

Amplification of CDK6 causes a marked increase of CDK6 expression and reduces the response of breast cancer cells to CDK4/6 inhibitors[34]. Interestingly, CDK6 has been shown to possess kinase-independent functions. For example, CDK6 acts as a chromatin-bound cofactor of p65 and associates with the promoters of nuclear factor-κB target genes, leading to the expression of genes associated with inflammation[35]. In addition, CDK6 forms a transcriptional complex with STAT3 or c-Jun to induce *p16INK4a* or *VEGF-A* expression in acute lymphoid leukemia, resulting in growth inhibition and angiogenesis, respectively[36]. However, the CDK4/6 inhibitor Palbociclib, similar to that of BET inhibitor, achieved a synergistic effect with AKTi in suppressing breast tumor cell growth, suggesting that the kinase activity of CDK6 is important for the intrinsic resistance to AKTi.

Lastly, our study provides a new mechanistic insight regarding FOXO3a-mediated gene transcription. Correlative analysis of FOXO3a and RNA polymerase II ChIP-seq profiles demonstrated that FOXO3a acts as a transcriptional activator through RNAPII recruitment[37]. Nevertheless, the mechanism of how FOXO3a recruits RNAPII and induces gene expression remains unclear. Acetylation is a major posttranslational modification of the FOXO3a protein. A study reported that acetylation is required for FOXO3a-induced expression of pro-apoptotic genes, such as Bim[23]. In addition, other studies found that FOXO3a acetylation is critical for its association with DNA sequence and transcriptional activity[38]. Our data demonstrated that K242/245 di-acetylation of FOXO3a gradually increases after prolonged AKTi treatment. Similar to Twist, this di-acetylation modification is located in a GK-X-GK motif, which can be recognized and bound by the second bromodomain of BRD4, as revealed by NMR structural and binding analysis. The K242/245 R mutant of FOXO3a loses its ability to interact with BRD4 and to induce CDK6 expression. Our study demonstrated that FOXO3a uses its di-acetylation motif to recruit BRD4 and RNA Polymerase II for activating gene transcription.

## Methods

**Cell culture**. All cell lines were purchased from ATCC. Breast cancer cell line BT474, MDA-MB-453, and MDA-MB-361 were grown in Dulbecco's modified Eagle's medium/F-12 medium plus 10% fetal bovine serum (FBS); T47D and ZR75 were grown in PRMI-1640 plus 10% FBS. MK2206-resistant cancer cells were obtained by stepwise increased concentrations of MK2206. BT474 and ZR75 cells were incubated with 1 μM MK2206 for 2 days in the beginning. Then the medium was changed to fresh one without MK2206 and cells were cultured until they grow well. Whenever we subcultured, the cells were incubated with gradual increasing concentration of MK2206 for 2 days. Some aliquots of the cells were stored whenever we subcultured it. Cells that grew at the maximum concentration (20 μM) of MK2206 were stored for further analyses. For selection of stable clone cells, puromycin (2 μg/ml) or blasticidin (5 μg/ml) was used.

**siRNA and antibodies**. Palbociclib, MK2206, AZD5363, and GSK690693 were from Selleckchem (Houston, TX). Smart pool siRNA (small interfering RNA) against human BRD4, BRD2, BRD3, and BRDT were obtained from Dharmacon (Chicago, IL). Short hairpin RNA against human FOXO3a and antibodies against Flag tag and actin were purchased from Sigma-Aldrich (St. Louis, MO). Antibodies for phospho-SirT6 (Ser338), SirT6, phospho-AKT substrate, AKT, BRD4, phospho-FOXO3a (Thr 32), FOXO3a, and Pan acetyl-lysine were purchased from Cell Signaling (Danvers, MA). Details regarding the dilutions of antibodies used were presented in Supplementary Table 1.

**Protein structure analysis by NMR**. NMR samples contained a protein/peptide complex of 0.5 mM in a 100 mM sodium phosphate buffer (pH 6.5) that contains 5 mM perdeuterated dithiothreitol and 0.5 mM EDTA in $H_2O/^2H_2O$ (9/1) or $^2H2O$. All NMR spectra were collected at 30 °C on NMR spectrometers of 800, 600, or 500 MHz. The $^1H$, $^{13}C$, and $^{15}N$ resonances of the protein in the complex were assigned by triple-resonance NMR spectra collected with a $^{13}C/^{15}N$-labeled and 75% deuterated BRD4-BD2 bound to an unlabeled FOXO3a peptide[39]. The distance restraints were obtained from 3D $^{13}C$-nuclear Overhauser spectroscopy (NOESY) or $^{15}N$-NOESY spectra. Protein structures were calculated with a distance geometry-simulated annealing protocol using X-PLOR that was aided by iterative automated NOE assignment by ARIA for refinement[40]. Structure quality was assessed by PROCHECK-NMR[41]. The structure of the protein/ligand complex was determined using intermolecular NOE-derived distance restraints that were obtained from $^{13}C$-edited ($F_1$), $^{13}C/^{15}N$-filtered ($F_3$) 3D NOESY spectra. Statistics of NMR structures of the BRD4-BD2/FOXO3a peptide complex are presented in Supplementary Table 2.

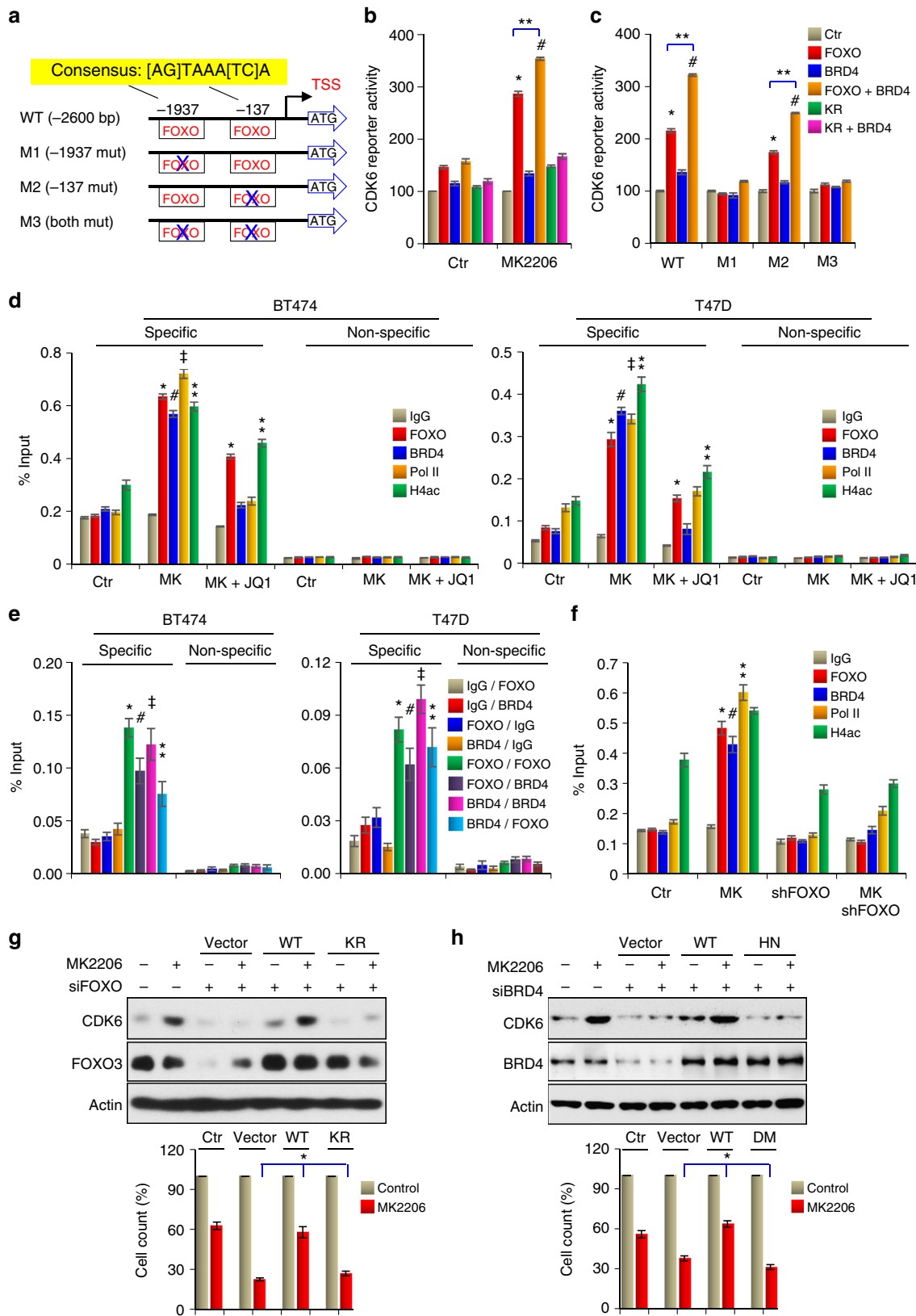

**Cell proliferation and MTT assay.** The cell proliferation was analyzed by counting the number of viable cells via Vi-CELL XR (Beckman Coulter). Breast cancer cells with the same number were seeded in culture dishes followed by specific treatment with different compounds for the indicated time, then trypsinized and counted the viable cells again. The cell proliferation rate was calculated by the difference of number of viable cells from two-times count. For MTT assay, seed cells in a 96-well plate at a density of $10^4$–$10^5$ cells/well in 100 µL of culture medium with compounds to be tested, culture the cells in a $CO_2$ incubator for indicated time. Then wash them by removing the medium and replacing it with 100 µL of fresh medium. Add 10 µL of the MTT stock solution to each well and

**Fig. 6** CDK6 is a targeted gene of FOXO3a/BRD4. **a** Schematic depiction of the *CDK6* promoter and *CDK6* reporter luciferase constructs used. Two consensus FOXO3a-binding motifs are indicated. **b** Effects of WT and KR mutant of FOXO3a, BRD4 on the activity of *CDK6* promoter luciferase in the presence or absence of MK2206. **c** Effects of WT-FOXO3a, BRD4 on the activation of WT or mutant *CDK6* promoter luciferase in the presence of MK2206. For **b** and **c**, */#/**$p < 0.01$. **d** FOXO3a, BRD4, H4ac4, and RNA-Pol II association at the *CDK6* promoter as assessed by ChIP. */*$p < 0.01$ when FOXO3a-ChIP or H4ac-ChIP in either MK2206-treated group or MK2206 plus JQ1-treated group is compared with control group. #/‡$p < 0.01$ when BRD4-ChIP or RNA-Pol II-ChIP in MK2206-treated group is compared with control group. **e** Cells treated with MK2206 were analyzed by sequential ChIP. The first-round of immunoprecipitation used anti-IgG, anti-FOXO3a or anti-BRD4 antibodies, followed by a second-round of immunoprecipitation using anti-IgG, anti-FOXO3a, or BRD4 antibodies. Pulldown DNA was analyzed by RT-PCR using specific *CDK6* promoter primers and nonspecific primers. */#/‡/*$p < 0.01$ when individual group with specific antibodies (FOXO3a and BRD4) used in the first-round of ChIP is compared with the corresponding group using IgG in the first-round of ChIP. **f** Effects of FOXO3a-knockdown, MK2206 treatment, or both on the association of FOXO3a, BRD4, RNA-Pol II, and H4ac at the CDK6 promoter as assessed by ChIP. */#/‡$p < 0.01$ when specific ChIP in MK2206-treated group is compared with the corresponding ChIP in control, or FOXO3a-knockdown or combination groups. **g** WT or KR mutant of FOXO3a was expressed in BT474 cells with endogenous FOXO3a knockdown. Rescue CDK6 expression was analyzed by western blotting; growth inhibition was assessed by cell count. *$p < 0.01$ when WT-FOXO3a group is compared with either vector or KR-FOXO3a group. **h** WT or HN mutant of BRD4 was expressed in BT474 cells with endogenous BRD4 knockdown. Rescue CDK6 expression and growth inhibition were assessed as in **g**. *$p < 0.01$ when WT-BRD4 group is compared with either vector or HN-BRD4 group. From **b** to **h**, data are representative of three experiments performed in triplicate as the mean ± SD

---

incubate them at 37 °C for 4 h. Add 100 μL of the SDS-HCl solution to each well, mix each sample by pipetting, and read absorbance at 570 nm. Statistical analysis (mean ± SD) from three separate experiments in triplicates is shown.

**In vitro deacetylation assay.** BT474 cells were treated with TSA (2 μM) and Nicotinamide (4 mM) for 12 h, and endogenous acetylated FOXO3a was pulled down by Anti-FOXO3a antibody from lysates. FOXO3a affinity beads were incubated with human recombinant active SirT6 protein in deacetylation buffer (25 mM Tris at pH 8.0, 137 mM NaCl, 2.7 mM KCl, and 1 mM MgCl$_2$) for 1 h at 30 °C. The acetylation status was analyzed by western blotting using pan acetyl-lysine antibody.

**Immunoprecipitation, immunoblotting, real-time PCR, immunofluorescence, and immunohistochemistry.** Experiments were performed as described previously[16]. In brief, cell lysates were extracted using IP buffer (50 mM Tris-HCl (pH7.4), 150 mM NaCl, 0.2 mM EDTA, 0.2% NP40, 10% Glycerol, and protease inhibitors) and immunoprecipitated with indicated antibodies and Protein G-Sepharose (Thermo). Pulldown protein complexes were analyzed by western blotting. For reverse-transcriptase PCR (RT-PCR) analysis, RNA was extracted from cells by RNeasy Mini Kit (#74104, Qiagen, Valencia, CA), and was reverse transcribed by SuperScriptR III Reverse Transcriptase (#18080044) from Thermo Fisher Scientific (Waltham, MA). Real-time PCR was analyzed using Power SYBR Green Master Mix (Applied Biosystems). The primers of *CDK6* gene for RT-PCR were: 5′- tgcacagtgtcacgaacaga-3′ and 5′- acctcggagaagctgaaaca-3′. For immuno-fluorescent staining, cells were grown on chamber slides, fixed with 4% paraformaldehyde, and incubated with primary antibodies. Secondary antibodies used were Alexa Fluor 488 goat anti-mouse IgG (H + L), Alexa Fluor 568 goat anti-mouse IgG (H + L), fluorescein isothiocyanate-conjugated goat anti-rabbit (Jackson ImmunoResearch Laboratories, West Grove, PA), or Alexa Fluor 350 goat anti-rabbit (Molecular Probe, Carlsbad, CA). For IHC, expression of CDK6 in human breast cancer tissue samples was stained with CDK6 antibody and control IgG, and each sample was scored by an H-score method that combines values of immunoreaction intensity with the percentage of cells staining.

**RNA-sequencing and data analysis.** Total RNA samples from MK2206-resistant and parental cells were extracted with PureLink RNA Mini Kit (Thermo Fisher), followed by on-column DNase digestion. NEBNext Poly(A) mRNA Magnetic Isolation Module (New England BioLabs) was used for polyA RNA purification. Libraries were prepared with NEBNext Ultra II Directional RNA Library Prep Kit for Illumina (New England BioLabs) and were sequenced to 75 bp paired-end on Illumina Hiseq 4000 platform. Sequenced reads were trimmed for adaptor sequence and then mapped to UCSC genome hg38 using Hisat2 (version 2.1.0)[42] with default parameters. Read counts were generated using featureCounts (versoin 1.6.0)[43] with "--ignoreDup -M -O --fraction -t exon" options and annotations from GENCODE Release 28 were used. The data have been deposited in the GEO repository with the accession number GSE118148.

**Chromatin immunoprecipitation.** ChIP assays were performed as described previously[16]. Approximately $1 \times 10^6$ BT474 and T47D cells were fixed with cross-link solution and collected, ChIP assays were performed using Imprint Chromatin Immunoprecipitation Kit (Sigma, #CHP1) according to the manufacturer's instructions. FOXO3a and BRD4 antibody-immunoprecipitated DNA was

analyzed by real-time PCR. Specific primers for the *CDK6* promoter were 5′-ACCT TCCCCTCCACGAGATA-3′ and 5′-GGGCGTGTGTTTAACTCCAA-3′. Unspecific primers for 3′-end region of *CDK6* gene as negative control of ChIP assay were 5′-TTGGGAAAGGGAGAACTGCA-3′ and 5′-AGCACCCAGTAA GACATCCA-3′. Samples were analyzed by real-time PCR using SYBR Green Power Master Mix following the manufacturer's protocol (Applied Biosystems). Sequential ChIP assay was performed using the Re-ChIP-IT magnetic chromatin reimmunoprecipitation kit (Active Motif, Carlsbad, CA) according to the manufacturer's protocol. Briefly, the chromatin–IgG, chromatin–FOXO3a, or chromatin–BRD4 complex was re-immunoprecipitated using anti-BRD4, anti-FOXO3a, or anti-IgG antibodies. After the Re-ChIP assay, the isolated DNA was analyzed through quantitative RT-PCR.

**Luciferase reporter assay.** HEK293 cells were seeded in 60 mm dishes and transfected with mentioned plasmids by FuGene 6 transfection reagent (Roche) for 24 h, cell lysates were extracted with passive lysis buffer (Promega, Madison, WI) and luciferase activity was measured using the Dual-Luciferase Reporter Assay System (Promega, Madison, WI). All experiments were performed for three times in triplicate. Representative experiment was shown.

**Flow cytometry.** Cell cycle was measured via flow cytometry analysis. Briefly, after specified treatment, cells were trypsinized and fixed with 70% ethanol for over an hour. Cells were then pelleted and washed with phosphate-buffered saline plus 20 mM EDTA. RNA was removed by incubating samples with RNase (1 mg/ml) at 37 °C for 2 h. Cells were then stained with propidium iodine (final concentration: 30 μg/ml), DNA content was analyzed by an LSR II flow cytometer (BD Biosciences). All experiments were performed for three times, statistical analysis (mean ± SD) from three separate experiments is shown.

**Mice model.** All procedures were approved by the Institutional Animal Care and Use Committee at the University of Kentucky College of Medicine and conform to the legal mandates and federal guidelines for the care and maintenance of laboratory animals. Animals were maintained and treated under pathogen-free conditions. Female athymic nude mice (6 weeks old; Taconic) were injected with breast cancer BT474 ($5 \times 10^6$ cells/mouse) cells subcutaneously[44,45]. When tumors reached 250 mm$^3$, the mice were divided into six groups (six mice/group) and received the following treatments: (1) control vehicle; (2) JQ1 (50 mg/kg); (3) Palbociclib (100 mg/kg); (4) MK2206 (90 mg/kg); (5) MK2206 plus JQ1; and (6) MK2206 plus Palbociclib every 2 days. Tumor growth was monitored with caliper measurements. When tumors were approximately 1.0 cm in size, mice were killed and tumors excised.

**Statistical analysis.** Data are presented as mean ± SD. Student's *t*-test (two-tailed) was used to compare two groups ($p < 0.05$ was considered significant) unless otherwise indicated.

## Data availability

RNA-sequencing data are available at the GEO data repository with the accession code GSE118148. The solution structure of the BRD4-BD2 in complex with Foxo3a-K242ac/K245ac peptide and the NMR spectral data are deposited in Protein Data Bank (PDB) ID 6MNL and BioMagResBank (BMRB) ID 30373,

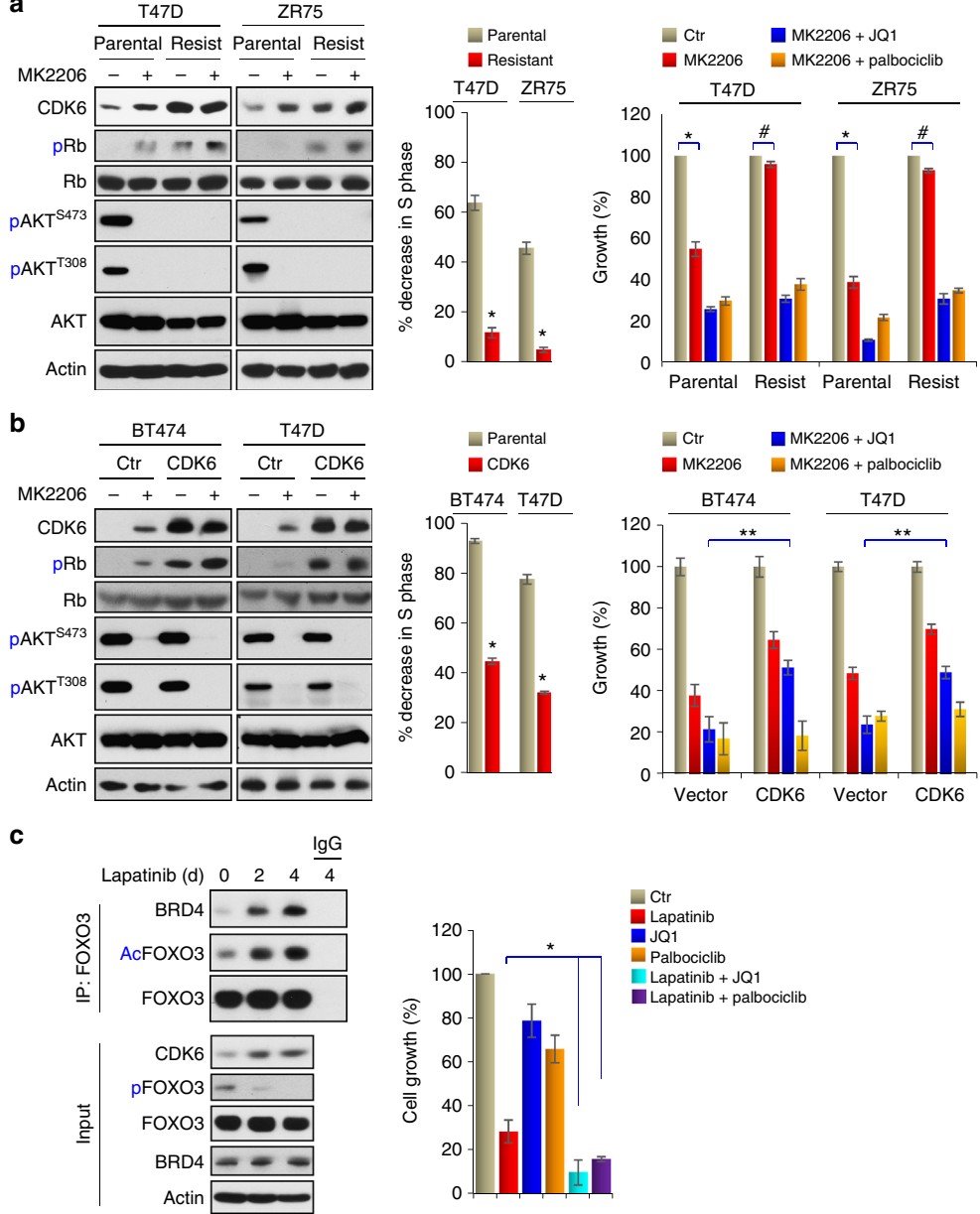

**Fig. 7** Inhibition of CDK6 confers drug sensitivity to AKTi. **a** Left panel: MK2206-resistant cell lines were established by growing T47D and ZR75 cells in increasing concentrations of MK2206 for 4 months. Inductions of CDK6 expression was examined by western blot. Middle panel: the growth suppressive effect of MK2206 was presented by the percentage of S-phase decrease in parental or resistant cells after 2 μM MK2206 treatment for 4 days. Data are representative of three experiments performed in triplicate as the mean ± SD. $^*p < 0.01$ when resistant cells were compared with parental cells. Right panel: the growth inhibition of these cells responded to the combination of MK2206 with JQ1 or Palbociclib was measured by cell count assays. Data are representative of three experiments performed in triplicate as the mean ± SD. $^*p < 0.01$ when MK2206-treated group was compared with untreated group in parental cells; $^\#p > 0.05$ when MK2206-treated group was compared with untreated group in resistant cells. **b** Left panel: induction of CDK6 expression in parental and CDK6-overexpressing cells were examined by western blotting. Middle panel: the growth suppressive effect of MK2206 was presented by the percentage of S-phase decreased in parental or CDK6-overexpressing cells after 2 μM MK2206 treatment for 4 days. Data are representative of three experiments performed in triplicate as the mean ± SD. $^*p < 0.01$ when CDK6-expressing cells were compared with vector control cells. Right panel: cell count assay was used to detect the growth inhibition effects of the combination of MK2206 with JQ1 or Palbociclib. Data are representative of three experiments performed in triplicate as the mean ± SD. $^{**}p < 0.01$ when MK2206 plus JQ1 treatment group in CDK6-expressing cells was compared with corresponding group in parental cells. **c** Lapatinib (1 μM) also induced FOXO3a acetylation, its association with BRD4 and CDK6 expression in BT474 cells as analyzed by IP-western blotting. The growth inhibition effects of the combination of Lapatinib with JQ1 or Palbociclib on BT474 cells were reflected by cell count assays. Data are representative of three experiments performed in triplicate as the mean ± SD. $^*p < 0.01$ when combinatory treatment group was compared with Lapatinib-treatment group

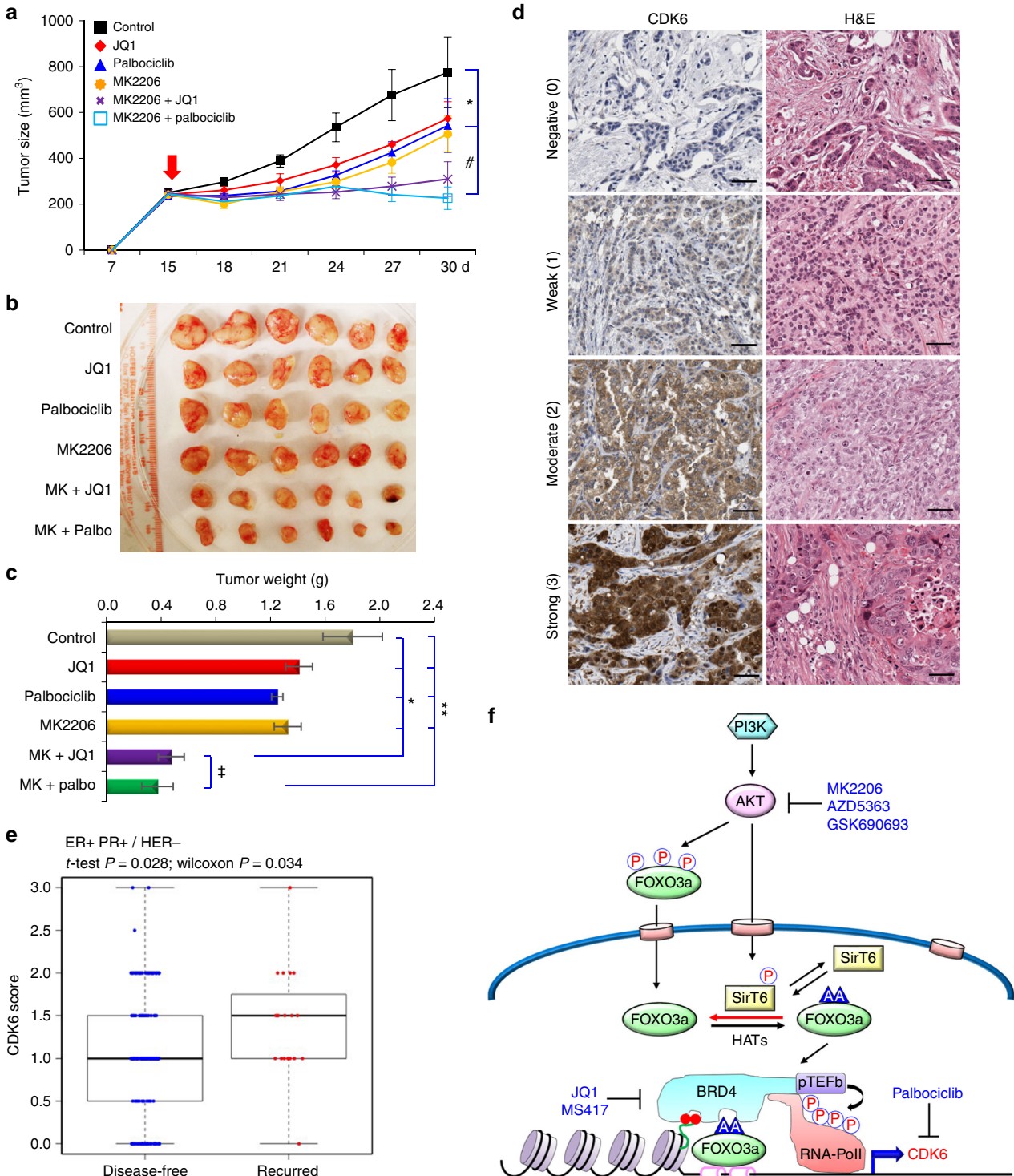

**Fig. 8** The FOXO3a-BRD4-CDK6 axis is critical for AKTi resistance in vivo. **a**, **b** BT474 cells were injected into athymic mice subcutaneously. When tumors from mice reached 250 mm$^3$, mice were divided into six groups (six mice/group) and treated with solvent control, MK2206 (90 mg/kg), JQ1 (50 mg/kg), Palbociclib (100 mg/kg), or in combinations, respectively. The size of tumor was recorded every 3 days. Data are represented as a mean ± SEM from six mice. */#$p < 0.01$ when combinatory treatment group is compared with either control or single-agent treatment group. **c** Tumor weight was also measured from above. Data are represented as a mean ± SEM from six mice. Data are represented as a mean ± SEM from six mice. */‡$p < 0.01$ when combinatory treatment group is compared with either control or single-agent treatment group. ‡$p > 0.05$ when both combinatory treatment groups were compared. **d**, **e** The 343 surgical specimens of breast cancer were immuno-stained using antibody against CDK6 and the control serum (data not shown). Images with consecutive IHC staining of CDK6 and H&E staining in four cases of breast tumors are shown (Scale bar = 100 μm). Recurrence rate was calculated according to CDK6 scores and clinical information of these patients. **f** A proposed model illustrating the formation of FOXO3a-BRD4 complex at the enhancer/promoter of CDK6, which leads to the transcriptional activation of CDK6 gene in response to AKTi. Either targeting FOXO3a-BRD4 complex by BET inhibitors or inhibition of CDK6 by Palbociclib reverses the AKTi resistance

respectively. All uncropped western blot images were presented in Supplementary Figure 6. All tumor images and source data of this study are available upon request.

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

## Acknowledgements

We thank Dr. Claudia Scholl for providing CDK6 expression plasmid. We also thank Dr. Cathy Anthony for critical reading and editing of this manuscript. This research was supported by the Shared Resources of the University of Kentucky Markey Cancer Center (P30CA177558) and University of Kentucky Center of Biomedical Research Excellence in Cancer and Metabolism (P20 GM121327). This work was also supported by grants from NIH (CA125454 and CA188118 to B.P.Z., and CA87658 and CA203067 to M.M.Z.), and DoD (BC140733 to M.M.Z. and B.P.Z.). This work was also supported in part by National Natural Science Foundation of China (81672629 to J.S.; 81402434 to Y.W.; 81530075 and 81773155 to S.L.), Science and Technology Program of Guangzhou, China (201707010331 to J.S.), and Basic Public Welfare Research Program of Zhejiang Province (LGF18H290003 to Y.W.).

## Author contributions

L.J., S.J., and Z.B.P. conceived and designed the study. L.J., S.J., D.Z., G.W., W.Y., C.Y., T. F., W.Y., L.Y., and H.Y. performed most of the study. Z.L., Z.Q., G.S., and Z.M.M. performed the NMR analyses and data interpretation. S.R.L. and L.S. performed the IHC and tumor sample analyses. W.C. performed statistical analyses. D.J., L.P.C., and E.B.M. discussed the results, conceived some experiments, and provided critical reagents and comments. L.J., L.S., S.J., Z.M.M., and Z.B.P. wrote the manuscript.

## Additional information

**Competing interests:** The authors declare no competing interests.

