## [Peer Review File · Nature Communications]

Reviewer #1 (Remarks to the Author):

MS#: NCOMMS-18-08917

Authors: Liu, Duan, Guo, Wu, Zeng, Zhang, Deng, Wang, Lin, Ghaffari, Evers, M Zhou, Shi, and B Zhou

In this manuscript, Liu et al. identified BRD4/FOXO3a-upregulated CDK6 promoter activity as a critical mechanism underlying resistance to AKT inhibition (AKTi; e.g., by MK2206) in several luminal breast cancer cells and thus blocking the action of BRD4 by JQ1 or MS417 and CDK6 by Palbociclib re-sensitizes AKTi-resistant cancer cells to growth inhibition and tumor suppression. Overall, the concept is intriguing and provides a molecular mechanism for AKTi drug resistance and re-sensitization in potential cancer treatment. However, there are some issues requiring authors' clarifications.

1. The emphasis on BRD4-FOXO3 complex-regulated CDK6 promoter activity needs to be substantiated, e.g., by sequential ChIP that is presently lacking. Although BRD4 interaction with FOXO3 is confirmed by co-IP, it remains uncertain whether the complex is indeed formed on the CDK6 promoter region.
2. Diacetylated FOXO3 interaction with BD2, but not BD1, of BRD4 needs to be demonstrated in the context of functional proteins. The study of protein fragment association by NMR, as presented, is encouraging but requires functional validation and structure-guided mutagenesis (e.g., H437 and N433) in the context of full-length proteins.
3. Whether the BRD4/FOXO/CDK6 axis is also operative in other types of breast cancer, besides luminal, needs to be addressed or discussed.

Technical comments:

4. In Figure 4, it is unclear whether SirT6 localization with WT and the TM mutant of FOXO3 has been examined and shown.
5. In Figure 5a, where dephosphorylation of FOXO3a occurs, either before or after nuclear entry, is unclear. In Figure 5b, enhancement of CDK6 upon MK2206 treatment in the cDNA microarray dataset is unconvincing. There appears to be no change. A heat map index for the color scale needs

to be shown. In Figure 5e, why not examine endogenous E2F targets? The abbreviation for PD needs to be defined.

6. In Figure 6c, the result seems to indicate the importance of distal, rather than proximal, FOXO-binding site as claimed in the text (page 9). Accordingly, the Specific Primers used in Figure 6d for either distal or proximal FOXO-binding site detection need to be clearly specified. In Figure 6e, the CDK6 signal seems decreased in BT474, but increased in T47D, in siBRD4 lanes without MK2206 treatment. Is this significant or simply experimental variation? In Figure 6f, is WT and KR FOXO3 interaction with BRD4 dependent on its BD2 and/or BD1?

7. In Figure 8, has orthotopic xenograft at mouse mammary fat pads that better reflect human breast cancer been attempted?

Editorial comments:

8. Besides MK2206 and AZD5363, the mechanistic action of GSK690693 should also be mentioned (page 5).

9. References for ATK phosphorylation of FOXO3 and its phosphorylation site mapping and localization changes should be cited in the text (page 5).

10. The abbreviation NLS (nuclear location sequence) mentioned in the text (page 6) is different from most of the terms defined in the literature.

11. The label NTC in Figure 1 needs to be defined.

12. Typos and grammar errors: two tandem bromodomain (page 6), SirT6 vs. SIRT6 (inconsistent usage throughout the manuscript), which inhibited by AKT (page 6), MK2066 (page 9), prolong treatment (page 11), high level of (page 13), (Mary Kay (page 18)

Reviewer #2 (Remarks to the Author):

This is a new study that proposes a role for the BRD4/FOXO3a axis in mediating resistance to AKT inhibition. Specifically, the authors propose that long term inhibition of AKT induces dephosphorylation of FOXO3, translocation to the nucleus and association with Sirt6 and recognition to BRD4. The net effect is transcriptional induction of CDK6 to the extent that inhibition of CDK6 or BRD4 sensitizes cells to AKT inhibition and overcoming resistance in vivo.

This is an interesting study with a good measure of novelty and is appropriate for NC. The data are by in large convincing, and indeed large! The finding of (yet) another mechanism of resistance to AKT inhibitors will be of interest to the field, and as far as I know, this is the first to invoke a role for combination with BRD4 inhibitors. I have a few questions, one pretty major one, for the authors to deal with before the paper can be considered for publication.

1. Fig. 1. None of these bar graphs have error bars or report on statistical significance.
2. Fig. 1C. What is 'cell growth (%)'. Why is this different from the cell count or MTT in 1A and 1B?
3. Fig. 2C. For this, and indeed many of the figures that use AKT inhibitors, there are no blots for pAKT, which is not really OK. All respectable studies using AKT pathway inhibitors have pAKT blots, and surrogate substrate readouts, which here FOXO3a is appropriate. Please add pAKT blots. S473 and T308.
4. Fig. 2d. this is a major point. Here they use prolonged AKT inhibition as a way of inducing the BRD4 pathway, as a result of dephosphorylated FOXO3a. But, in numerous studies, prolonged inhibition with AKT inhibitors results in rebound reactivation of AKT (papers by Rosen lab, and others), which would presumably result in re-phosphorylation of FOXO3. Why isn't there rebound AKT activation, which after all is the mechanism of emergence of resistance to AKT inhibitors? Again, there are no pAKT blots, just a pFOXO3a blot. Do other AKT inhibitors also show this? The lack of rebound AKT activation needs to be explored.

5. Fig, 4f. the last lane in each panel is presumably IgG control? This is not labeled.
6. Fig. 6 again no error bars, no stats on any of these bar graphs,
7. Fig 7a. Again, no AKT or pAKT blots here, and these are critical.
8. Fig, 8a. are these effects significant? I don't see any p values.

Point-By-Point Response to the Reviewers' Comments

We are pleased that the reviewers found our study to be novel and important, and also appreciative of their helpful and constructive comments. We have taken the comments from all reviewers seriously and revised our manuscript extensively. We believe that with our new data, the revision has dramatically strengthened our study and addressed the reviewers' concerns. Below, we respond to the comments made by each reviewer.

New data and changes in Figures

1. Sequential ChIP analysis of the binding of BRD4 and FOXO3 at the *CDK6* promoter (Fig 6e)
2. FOXO3-knockdown blocks the association of BRD4 at the *CDK6* promoter (Fig 6f)
3. FOXO3 interacts with wild-type BRD4, but not the His437/Asn433 double-mutant of BRD4 (Fig 3f)
4. Expression of WT- and His437/Asn433 double-mutant BRD4 in BT474 cells with BRD4-KD (Fig 6h)
5. RNA sequencing analysis in two MK2206-treated luminal breast cancer cell lines (Fig 5b and Fig S3)
6. Detection of AKT-S473 & -T308 phosphorylation in Figs 2c, 2d, 2e, 2f, 4c, 4f, 4g, 7a, and 7b
7. Immunofluorescence staining detects the co-localization of WT- or TM-FOXO3 with SIRT6 (Fig S2)
8. MK2206 does not induce *CDK6* upregulation in basal-like breast cancer (BLBC) cell lines (Fig S4a)
9. Repeat experiment of Fig 6f (original submission) and the new data is presented on Fig S4b

Reviewer #1

(Remarks to the Author):

In this manuscript, Liu et al. identified BRD4/FOXO3a-upregulated CDK6 promoter activity as a critical mechanism underlying resistance to AKT inhibition (AKTi; e.g., by MK2206) in several luminal breast cancer cells and thus blocking the action of BRD4 by JQ1 or MS417 and CDK6 by Palbociclib re-sensitizes AKTi-resistant cancer cells to growth inhibition and tumor suppression. Overall, the concept is intriguing and provides a molecular mechanism for AKTi drug resistance and re-sensitization in potential cancer treatment.

Response: We appreciate the positive comment from Reviewer#1.

Major comments:

1. *The emphasis on BRD4-FOXO3 complex-regulated CDK6 promoter activity needs to be substantiated, e.g., by sequential ChIP that is presently lacking. Although BRD4 interaction with FOXO3 is confirmed by co-IP, it remains uncertain whether the complex is indeed formed on the CDK6 promoter region.*

Response: We appreciate the insightful comment from Reviewer#1. As suggested, we have performed the sequential ChIP analyses to further validate the association of the FOXO3a-BRD4 complex at the *CDK6* promoter (**Fig. 6e**). When using a control IgG in the first cycle of ChIP, we did not find the association of FOXO3a or BRD4 at the *CDK6* promoter in the sequential ChIP. However, using either FOXO3a or BRD4 antibody in the first cycle of ChIP, we found the interaction of BRD4 or FOXO3a at the *CDK6* promoter (**Fig. 6e**). In addition, we knocked down FOXO3a in BT474 cells and found that FOXO3a-knockdown significantly reduced BRD4 association at the *CDK6* promoter (**Fig 6f**). Together, these data further strengthen our conclusion that the FOXO3a-BRD4 interaction is critical for the transcription of *CDK6*.

2. *Diacetylated FOXO3 interaction with BD2, but not BD1, of BRD4 needs to be demonstrated in the context of functional proteins. The study of protein fragment association by NMR, as presented, is encouraging but requires functional validation and structure-guided mutagenesis (e.g., H437 and N433) in the context of full-length proteins.*

Response: We appreciate this illuminating comment from Reviewer#1. We have generated a double mutant full-length BRD4 (HN), in which both His437 and Asn433 were changed to Alanine. HN-BRD4 totally lost the ability to bind FOXO3a (**Fig 3f**). In addition, when WT- and HN-BRD4 were expressed in BRD4-knockdown BT474 cells, we found that WT-BRD4 restored *CDK6* expression. However, HN-BRD4 could not rescue *CDK6* expression under MK2206 stimulation (**Fig 6h**). Together, these data support our structural NMR analysis that His437 and Asn433 in BD2 of BRD4 are critical for the binding of FOXO3a in *CDK6* transcription.

3. *Whether the BRD4/FOXO/CDK6 axis is also operative in other types of breast cancer, besides luminal, needs to be addressed or discussed.*

Response: We appreciate the constructive comment from Reviewer#1. Besides luminal subtype of breast cancer, basal-like breast cancer (BLBC), commonly referred as triple-negative breast cancer (TNBC), has been reported to have *CDK6* overexpression¹. To assess whether the BRD4/FOXO3a/*CDK6* axis is also involved in

CDK6 transcription in BLBC, we treated two BLBC cell lines, MDA-MB-231 and BT549, with MK2206 for up to 4 days (**Supplemental Fig S4a**). However, we did not find CDK6 upregulation in BLBC, suggesting that the BRD4/FOXO3a/CDK6 signaling axis is only operative in luminal subtype of breast cancer. This is consistent with the notion that different breast cancer subtypes have different genetic makeups^{2, 3, 4}.

Technical comments:

4. In Figure 4, it is unclear whether SirT6 localization with WT and the TM mutant of FOXO3 has been examined and shown.

Response: We greatly appreciate this constructive comment from Reviewer#1. We expressed exogenous WT- and the TM-mutant of FOXO3a (HA-tagged) in cells, and performed an immunofluorescence staining to examine the cellular localizations of FOXO3a (using mouse HA antibody) and endogenous SirT6 (using rabbit SirT6 antibody) in these cells (**Supplemental Fig S2**). We found that WT-FOXO3 is mainly localized in the cytoplasm; MK2206 treatment induced the nuclear localization of WT-FOXO3a. However, TM-FOXO3a, in which three AKT-phosphorylation sites were mutated to Alanine, resided exclusively in the nucleus. This finding is consistent with previous reports that AKT-mediated FOXO3a phosphorylation is essential to retain the cytoplasmic localization of FOXO3a^{5, 6}. We also found that endogenous SirT6 resided in the nucleus and its nuclear localization did not change after MK2206 treatment.

5. In Figure 5a, where dephosphorylation of FOXO3a occurs, either before or after nuclear entry, is unclear. In Figure 5b, enhancement of CDK6 upon MK2206 treatment in the cDNA microarray dataset is unconvincing. There appears to be no change. A heat map index for the color scale needs to be shown. In Figure 5e, why not examine endogenous E2F targets? The abbreviation for PD needs to be defined.

Response: We appreciate illuminating comments from Reviewer#1. AKT-mediated FOXO3a phosphorylation is the major mechanism responsible for the cytoplasmic retention of FOXO3a. When AKT is inhibited, FOXO3a phosphorylation is reduced by the protein phosphatase 2A (PP2A)⁷, which is constitutively active in the cytoplasm. This reduced phosphorylation leads to the dis-association of 14-3-3 and the consequent exposure of the FOXO3a nuclear localization signal (NLS), which results in the nuclear translocation of FOXO3a. For the clarity of this study, we did not provide the aforementioned details on Fig 5a. However, we have included a description and reference on the revised manuscript (please see lane 176 in page 7).

We apologize for the low resolution of Figure 5b, which was based on cDNA microarray analysis. We have repeated this experiment by using RNA-seq analysis. The RNA-seq data have been deposited in GEO with an access number GSE118148. Based on the RNA-seq analysis, differentially expressed genes with more than 2.5-fold difference from both BT474 and T47D cells were identified to compile to the current heat-map (**Fig 5b**). Consistent with previous cDNA microarray analysis, CDK6 was one of the top genes with more than 2.5-fold upregulation after MK2206 treatment in both BT474 and T47D cells.

CDK6 is well-recognized to phosphorylate RB and activate E2F expression, we identified several E2F target genes (MYC, CDKN1A, CDKN1B, HMGA1, SMC3, and NAP1L1) that were also upregulated in MK2206-treated cells (GSE118148). Intriguingly, numerous recent studies demonstrated that CDK6 has other cell-cycle independent functions that do not require E2F. For example, CDK6 phosphorylates Forkhead Box M1 (FOXO1) to suppress cellular senescence in tumor cells⁸. CDK6 also phosphorylates 6-phosphofruktokinase (PFK1) and pyruvate kinase M2 (PKM2) to rewire the metabolic program in cancer cells⁹. In addition, CDK6 suppresses anti-tumor immunity^{10, 11}. It is unclear whether CDK6-mediated drug resistance requires a cell-cycle dependent or independent function. Future systematic analyses will yield comprehensive understanding on this interesting area.

PD stands for Palbociclib (PD0332991); we have defined this abbreviation in the figure legend of Fig 5e.

6. In Figure 6c, the result seems to indicate the importance of distal, rather than proximal, FOXO-binding site as claimed in the text (page 9). Accordingly, the Specific Primers used in Figure 6d for either distal or proximal FOXO-binding site detection need to be clearly specified. In Figure 6e, the CDK6 signal seems decreased in BT474, but increased in T47D, in siBRD4 lanes without MK2206 treatment. Is this significant or simply experimental variation? In Figure 6f, is WT and KR FOXO3 interaction with BRD4 dependent on its BD2 and/or BD1?

Response: We apologize for our mistake on Page 9; the 'distal' FOXO3-binding site on the CDK6 promoter is responsible for FOXO3 recognition. We have corrected this and supplied the primer sequences in the Materials and Methods section on the revised manuscript (please see lane 253 in page 10 and page 18).

We also repeated the experiments presented in Figure 6E of the original submission, confirmed that CDK6

expression was decreased in both BT474 and T47D cells with BRD4-knockdown. This new data is presented in Supplemental Fig. S4b of the revised manuscript.

Similar to our response for comment#2, we found that HN-BRD4 (His437/Asn433 changed to Alanine), which cannot interact with FOXO3a (Fig 3f), failed to restore CDK6 expression compared to WT-BRD4 (Fig 6h). These findings indicate that the FOXO3a-BRD4 interaction is critical for CDK6 transcription.

7. In Figure 8, has orthotopic xenograft at mouse mammary fat pads that better reflect human breast cancer been attempted?

Response: We thank Reviewer#1 for this interesting question. For unknown reasons, BT474 cells are commonly implanted subcutaneously to generate the breast xenograft model^{12, 13, 14, 15}. To clarify this, we have included these references on the Materials and Methods section in our revised manuscript.

Editorial comments:

8. Besides MK2206 and AZD5363, the mechanistic action of GSK690693 should also be mentioned (page 5).

Response: We appreciate the constructive comment from Reviewer#1. We have included the brief introduction of GSK690693 and corresponding reference into manuscript (please see line 100 in page 5).

9. References for ATK phosphorylation of FOXO3 and its phosphorylation site mapping and localization changes should be cited in the text (page 5).

Response: We appreciate the constructive comment from Reviewer#1. We have added the following two original references in our revised manuscript (please see line 113 in page 5).

(1) Brunet, A. et al. Akt promotes cell survival by phosphorylating and inhibiting a forkhead transcription factor. *Cell* 96, 857–868 (1999).

(2) Kops, G. J. P. L. et al. Direct control of the forkhead transcription factor AFX by protein kinase B. *Nature* 398, 630–634 (1999).

10. The abbreviation NLS (nuclear location sequence) mentioned in the text (page 6) is different from most of the terms defined in the literature.

Response: We appreciate the constructive comment from Reviewer#1. We have corrected the term to 'nuclear localization signal' on our revised manuscript (please see line 118 in page 6).

11. The label NTC in Figure 1 needs to be defined.

Response: We appreciate the constructive comment from Reviewer #1. We have defined the meaning of 'NTC' in the figure legend on our revised manuscript (please see page 24).

12. Typos and grammar errors: two tandem bromodomain (page 6), SirT6 vs. SIRT6 (inconsistent usage throughout the manuscript), which inhibited by AKT (page 6), MK2066 (page 9), prolong treatment (page 11), high level of (page 13), (Mary Kay (page 18)).

Response: We appreciate the constructive comment from Reviewer #1. We have carefully corrected these typos and grammar errors on our revised manuscript.

Reviewer #2:

(Remarks to the Author):

This is a new study that proposes a role for the BRD4/FOXO3a axis in mediating resistance to AKT inhibition. Specifically, the authors propose that long term inhibition of AKT induces dephosphorylation of FOXO3, translocation to the nucleus and association with Sirt6 and recognition to BRD4. The net effect is transcriptional induction of CDK6 to the extent that inhibition of CDK6 or BRD4 sensitizes cells to AKT inhibition and overcoming resistance in vivo.

This is an interesting study with a good measure of novelty and is appropriate for NC. The data are by in large convincing, and indeed large! The finding of (yet) another mechanism of resistance to AKT inhibitors will be of interest to the field, and as far as I know, this is the first to invoke a role for combination with BRD4 inhibitors. I have a few questions, one pretty major one, for the authors to deal with before the paper can be considered for publication.

Response: We greatly appreciate the overall appraisal of our study by Reviewer#2

Specific comments

1. Fig 1. None of these bar graphs have error bars or report on statistical significance.

Response: We appreciate the constructive comment from Reviewer#2. We have carefully modified these data and included the report of statistical significance.

2. Fig 1C. What is 'cell growth (%)'. Why is this different from the cell count or MTT in 1A and 1B?

Response: We appreciate the constructive comment from Reviewer#2. We have corrected it to 'cell count', which is consistent with Fig1A.

3. Fig 2C. For this, and indeed many of the figures that use AKT inhibitors, there are no blots for pAKT, which is not really OK. All respectable studies using AKT pathway inhibitors have pAKT blots, and surrogate substrate readouts, which here FOXO3a is appropriate. Please add pAKT blots: S473 and T308.

Response: We appreciate the illuminating comment from Reviewer#2. We have detected both AKT-S473 and T308 phosphorylation status in Figs 2c, 2d, 2e, 2f, 4c, 4f, 4g, 7a, and 7b.

4. Fig 2d. this is a major point. Here they use prolonged AKT inhibition as a way of inducing the BRD4 pathway, as a result of dephosphorylated FOXO3a. But, in numerous studies, prolonged inhibition with AKT inhibitors results in rebound reactivation of AKT (papers by Rosen lab, and others), which would presumably result in re-phosphorylation of FOXO3. Why isn't there rebound AKT activation, which after all is the mechanism of emergence of resistance to AKT inhibitors? Again, there are no pAKT blots, just a pFOXO3a blot. Do other AKT inhibitors also show this? The lack of rebound AKT activation needs to be explored.

Response: We appreciate the illuminating comment from Reviewer#2. As elegantly shown by Dr. Rosen and his colleagues¹², a partially rebound in AKT phosphorylation occurs under prolonged AKT inhibition. We have measured both AKT-S473 and T308 phosphorylation and obtained similar results (**Fig 2d**). However, the minor rebound in AKT phosphorylation is not likely the mechanism responsible for AKTi resistance based on the following evidence. First, AKT is phosphorylated by PDK1 and mTORC2 at T308 and S473, respectively. However, AKTi does not inhibit PDK1 and mTORC2 directly. MK2206 suppresses the allosteric activation of AKT, whereas AZD5363 and GSK690693 directly bind to the catalytic core of AKT and inhibit its enzymatic activity. Therefore, the minor rebound in AKT phosphorylation after prolonged AKTi treatment is due to a feedback mechanism, and does not suggest AKT re-activation. Consistent with this idea, AZD5363 and GSK690693 blocked activation of AKT targets (such as FOXOs, S6K and GSK3 β)¹⁶ without affecting AKT phosphorylation^{17, 18}.

Second, although we also observed a minor rebound in AKT phosphorylation after prolonged AKT inhibition, FOXO3a, the major downstream substrate of AKT, remained un-phosphorylated (as shown in **Figs 2c, 2d, 2e, 2f** and **4c**). Similarly, in Dr. Rosen's study, S6K, also known to be an AKT target, remains completely un-phosphorylated despite a rebound AKT-phosphorylation [Fig 1A in *Cancer Cell*, 2011; 19(1):58-71]¹². These observations indicate that the minor rebound feedback AKT phosphorylation does not represent AKT re-activation and is not likely responsible for the resistance to AKTi.

Third, we found that the FOXO3/BRD4 complex is critical for CDK6 transcription, which occurs in 24 hr and continues to increase after 4-day of MK2206 treatment (**Fig 5d**). Consistent with this, FOXO3a remained un-phosphorylated during the entire period of 4-day of MK2206 treatment despite a minor rebound of AKT phosphorylation (**Fig 2d**; please note lysates are the same for Fig 2d and 5d). In addition, exogenous CDK6 expression can block the sensitivity of cells to AKTi (**Fig 7d**). Together, these findings indicate that CDK6 induction, mediated by the FOXO3a/BRD4 complex, is one of the mechanisms responsible for AKTi resistance.

5. Fig, 4f. the last lane in each panel is presumably IgG control? This is not labeled.

Response: They were indeed the lanes of IgG control and we have added the label into Figure 4F.

6. Fig. 6 again no error bars, no stats on any of these bar graphs.

Response: We have modified these data and included the *p* values.

7. Fig 7a. Again, no AKT or pAKT blots here, and these are critical.

Response: We have included AKT phosphorylation (T308 & S473) in Fig 7a on our revised manuscript.

8. Fig, 8a. are these effects significant? I don't see any *p* values.

Response: We have modified these data and included the *p* values.

References:

1. Hsu YH, *et al.* Definition of PKC-alpha, CDK6, and MET as therapeutic targets in triple-negative breast cancer. *Cancer Res* **74**, 4822-4835 (2014).
2. Banerji S, *et al.* Sequence analysis of mutations and translocations across breast cancer subtypes. *Nature* **486**, 405-409 (2012).
3. Cancer Genome Atlas N. Comprehensive molecular portraits of human breast tumours. *Nature* **490**, 61-70 (2012).
4. Stephens PJ, *et al.* The landscape of cancer genes and mutational processes in breast cancer. *Nature* **486**, 400-404 (2012).
5. Brunet A, *et al.* Akt promotes cell survival by phosphorylating and inhibiting a Forkhead transcription factor. *Cell* **96**, 857-868 (1999).
6. Kops GJ, de Ruiter ND, De Vries-Smits AM, Powell DR, Bos JL, Burgering BM. Direct control of the Forkhead transcription factor AFX by protein kinase B. *Nature* **398**, 630-634 (1999).
7. Singh A, *et al.* Protein phosphatase 2A reactivates FOXO3a through a dynamic interplay with 14-3-3 and AKT. *Molecular biology of the cell* **21**, 1140-1152 (2010).
8. Anders L, *et al.* A systematic screen for CDK4/6 substrates links FOXM1 phosphorylation to senescence suppression in cancer cells. *Cancer Cell* **20**, 620-634 (2011).
9. Wang H, *et al.* The metabolic function of cyclin D3-CDK6 kinase in cancer cell survival. *Nature* **546**, 426-430 (2017).
10. Goel S, *et al.* CDK4/6 inhibition triggers anti-tumour immunity. *Nature* **548**, 471-475 (2017).
11. Zhang J, *et al.* Cyclin D-CDK4 kinase destabilizes PD-L1 via cullin 3-SPOP to control cancer immune surveillance. *Nature* **553**, 91-95 (2018).
12. Chandarlapaty S, *et al.* AKT inhibition relieves feedback suppression of receptor tyrosine kinase expression and activity. *Cancer Cell* **19**, 58-71 (2011).
13. Rodrik-Outmezguine VS, *et al.* mTOR kinase inhibition causes feedback-dependent biphasic regulation of AKT signaling. *Cancer Discov* **1**, 248-259 (2011).
14. Schwartz S, *et al.* Feedback suppression of PI3Kalpha signaling in PTEN-mutated tumors is relieved by selective inhibition of PI3Kbeta. *Cancer Cell* **27**, 109-122 (2015).
15. Will M, *et al.* Rapid induction of apoptosis by PI3K inhibitors is dependent upon their transient inhibition of RAS-ERK signaling. *Cancer Discov* **4**, 334-347 (2014).
16. Manning BD, Toker A. AKT/PKB Signaling: Navigating the Network. *Cell* **169**, 381-405 (2017).
17. Davies BR, *et al.* Preclinical pharmacology of AZD5363, an inhibitor of AKT: pharmacodynamics, antitumor activity, and correlation of monotherapy activity with genetic background. *Mol Cancer Ther* **11**, 873-887 (2012).
18. Rhodes N, *et al.* Characterization of an Akt kinase inhibitor with potent pharmacodynamic and antitumor activity. *Cancer Res* **68**, 2366-2374 (2008).

Reviewer #1 (Remarks to the Author):

MS#: NCOMMS-18-08917A

Authors: Liu, Duan, Guo, Wu, Chen, Y Wang, Y Lin, Zeng, Zhang, Deng, Stewart, C Wang, P Lin, Ghaffari, Evers, M Zhou, Shi, and B Zhou

In this revision, the authors have properly addressed my previous comments. The only suggestion is to align better, in Figure 6g, the - and + labels on the top two rows with the gel images. This revised manuscript is recommended for publication in Nature Communications.

Reviewer #2 (Remarks to the Author):

None

Point-By-Point Response to Reviewer' Comments and Editorial Requests

We thank reviewers' constructive comments and their recommendation for publication of this study.

Minor change from Reviewer#1:

"In this revision, the authors have properly addressed my previous comments. The only suggestion is to align better, in Figure 6g, the - and + labels on the top two rows with the gel images. This revised manuscript is recommended for publication in Nature Communications."

Response: We greatly appreciate the suggestion from Reviewer#1, and we have made this correction.

Editorial Requests

- (1) We agreed to publish reviewers' comments and our rebuttal letter.
- (2) We have made changes by strictly following the editorial guidelines; these changes are outlined below. An updated editorial policy checklist is submitted with this manuscript.

Data Availability:

- We have added this paragraph after the Methods section. RNA-sequencing data are available at the GEO data repository with the accession code GSE118148. The solution structure of the BRD4-BD2 in complex with Foxo3a-K242ac/K245ac peptide and the NMR spectral data are deposited in Protein Data Bank (PDB) ID 6MNL, and BioMagResBank (BMRB) ID 30373, respectively.

Methods:

- Animal experiments are approved by our institutional IACUC Committee and compiled with ethical regulations.
- All cell lines are purchased from ATCC and no misidentified cell lines are used.
- A detailed table (Supplementary Table 1) is provided to list all the commercial antibodies used, including the catalogue numbers, applications, and the dilution used.
- We have provided all the uncropped scans of western blot images in the Supplementary Figure 6. Molecular weight markers are also listed in these images.
- Error bars are clearly presented and p-values are stated in the corresponding Figure legend.
- Color scales are clearly defined in Figures.
- All figure legends do not exceed 350 words.
- We have provided "Author Contributions" section after the acknowledgement.
- A separate and complete Supplementary Information is provided, including 6 Supplementary Figures and 2 Supplementary Tables (in one single PDF file)
- We have carefully checked our manuscript and provided relevant citation numbers throughout text.

Editorial Summary

The editorial summary is excellent and succinctly presented.